# Perceptual decisions interfere more with eye movements than with reach movements

Kazumichi Matsumiya [1✉] & Shota Furukawa[1]

Perceptual judgements are formed through invisible cognitive processes. Reading out these judgements is essential for advancing our understanding of decision making and requires inferring covert cognitive states based on overt motor actions. Although intuition suggests that these actions must be related to the formation of decisions about where to move body parts, actions have been reported to be influenced by perceptual judgements even when the action is irrelevant to the perceptual judgement. However, despite performing multiple actions in our daily lives, how perceptual judgements influence multiple judgement-irrelevant actions is unknown. Here we show that perceptual judgements affect only saccadic eye movements when simultaneous judgement-irrelevant saccades and reaches are made, demonstrating that perceptual judgement-related signals continuously flow into the oculo-motor system alone when multiple judgement-irrelevant actions are performed. This suggests that saccades are useful for making inferences about covert perceptual decisions, even when the actions are not tied to decision making.

[1] Graduate School of Information Sciences, Tohoku University, Sendai, Japan. ✉email: matsumiya@tohoku.ac.jp

Studies of perceptual judgements depend on the ability to make inferences about covert cognitive states. To infer such covert cognitive states, overt motor actions are commonly used. Perceptual judgements and motor actions are often modelled as serial stages of processing. In a perceptual judgement task, it is often assumed that first a perceptual judgement is completed and then the subsequent motor output is planned and executed[1,2]. For example, saccadic eye movements are made after a decision about where to move the eyes based on sensory information.

However, the accumulated literature indicates that motor actions are continuously affected by ongoing perceptual judgement processes that are not yet complete[3,4], suggesting an interaction between perceptual judgements and motor actions. In a variety of reach movement tasks, the trajectories of reach movements have been shown to be modulated by a target selection process in visual search[5], a lexical decision process[6] and the magnitude of a single Arabic numeral[7,8]. These findings indicate that reach movements are not always the final product of perceptual judgements and that ongoing perceptual judgements continuously affect reach movements, suggesting a continuous interaction between perceptual judgements and reach movements.

Furthermore, the trajectories of saccadic eye movements also elicit systematic deviations in saccade curvature and endpoints when saccadic eye movements are used to report judgements in a perceptual judgement task[9], indicating that oculomotor output can also be continuously affected by ongoing perceptual judgements. A recent study has demonstrated that saccades are not yet ready to launch when perceptual decision processes terminate[10], suggesting that perceptual decisions and oculomotor responses rely on temporally distinct streams of evidence. These findings imply a continuous interaction between perceptual judgements and saccadic eye movements[11], like reach movements.

Continuous interactions between perceptual judgements and motor actions may be based on interference of signals in neural circuits. Neural responses in oculomotor brain circuits (e.g., the lateral intraparietal area [LIP]) have been reported to show heterogeneous selectivity for different sources, such as the formation of perceptual judgements and the execution of eye movements, within the same neurons[12–16]. Neurons in manual brain circuits (e.g., the medial intraparietal area [MIP]) have also been reported to show selectivity for both the formation of perceptual judgements and the execution of reach movements[17]. Thus, the interference of signals related to the formation of perceptual judgements and motor execution in motor brain areas seems to provide a neural basis by which these multiple signals can continuously interact with each other.

Interestingly, interference of perceptual judgement-related signals and judgement-irrelevant saccade responses has also been observed in the LIP of monkeys[12–16]. This neurophysiological observation has been supported by a recent human behavioural study in which the formation of perceptual judgements affected saccadic eye movements, even when the saccadic eye movements were irrelevant to the perceptual judgement task[18]. Thus, the effects of perceptual judgements on judgement-irrelevant motor actions may be considered a side effect of signal interference in motor brain areas.

However, it is not known how the signal interference occurs in dual-task paradigms such as simultaneous eye and reach movements. Such paradigms offer the opportunity to investigate how perceptual judgement-related signals flow between motor systems, which helps to explain the mechanisms of communication across motor systems[19] during interference between perceptual decision making and motor actions. A previous neurophysiological study showed that perceptual judgement-related activity

arises in both oculomotor and manual brain areas such as the LIP and MIP, respectively[17]. Importantly, that neurophysiological study also showed that the activity of MIP neurons is greatly attenuated when perceptual judgements are communicated by eye movements (i.e., due to reduced perceptual judgement-related signals from MIP neurons, these signals likely interfere less with reach movements when eye movements are made) while LIP neurons still activate when perceptual judgements are communicated by reach movements, as well as eye movements (i.e., perceptual judgement-related signals from LIP neurons can still interfere with eye movements when reach movements are made)[17]. Therefore, we hypothesised that perceptual judgements may interfere more with eye movements than with reach movements when simultaneous judgement-irrelevant eye and reach movements are made (hypothesis 1).

Furthermore, it is not known whether perceptual judgements interfere with judgement-irrelevant reach movements without eye movements. Perceptual judgement-related activity arises in the LIP[1,12,17,20], and such activity interferes with judgement-irrelevant eye movements[18]. Given that the MIP shows selectivity for both perceptual judgement-related activity and motor processes, similar to the LIP[17], we hypothesised that perceptual judgement-related activity might interfere with judgement-irrelevant reach movements when reach movements are made without eye movements (hypothesis 2).

If we can obtain results that support hypotheses 1 and 2, these results would demonstrate that perceptual judgement-related signals continuously flow into the oculomotor system alone when multiple judgement-irrelevant actions are performed. Testing of hypothesis 1 would reveal whether simultaneous eye and reach movements are necessary for perceptual judgements to interfere only with judgement-irrelevant eye movements. However, even if we obtain results that support hypothesis 1, perceptual judgements might not interfere with judgement-irrelevant reach movements regardless of simultaneous eye and reach movements. To address this issue, we need to test hypothesis 2. Testing of hypothesis 2 would reveal whether interference between perceptual judgements and judgement-irrelevant motor actions is observed in reach movements without eye movements. Therefore, the fact that both hypotheses 1 and 2 are true would suggest that perceptual judgements interfere more with eye movements than with reach movements when simultaneous judgement-irrelevant eye and reach movements are made. These results will provide clues for understanding the mechanisms of communication across motor systems during perceptual decision making[19].

To test the two hypotheses, we developed a paradigm in which eye and reach movements had to be made simultaneously but were independent of a concurrent perceptual judgement task. We then looked for perturbations in the reaction times and peak velocities of eye or reach movements. The paradigm used in the present study differed from previous paradigms. For example, in the previous paradigms, participants discriminated changes in visual targets during eye movements[21–24] or reach movements[25]. In other examples, participants discriminated briefly presented visual patterns during eye and reach movements[26,27] or judged the location of body parts during eye and reach movements[28,29]. Thus, the previous paradigms assessed how eye and reach movements affect concurrent perceptual processes in which participants performed a perceptual task during eye and reach movements[21–29]. We reversed the logical order with the aim of examining how perceptual processes affect concurrent eye and reach movements and measured saccade and reach metrics during an ongoing perceptual judgement task (Fig. 1). Participants were first asked to judge the direction of a visual motion stimulus that was briefly presented on the display and later to respond by

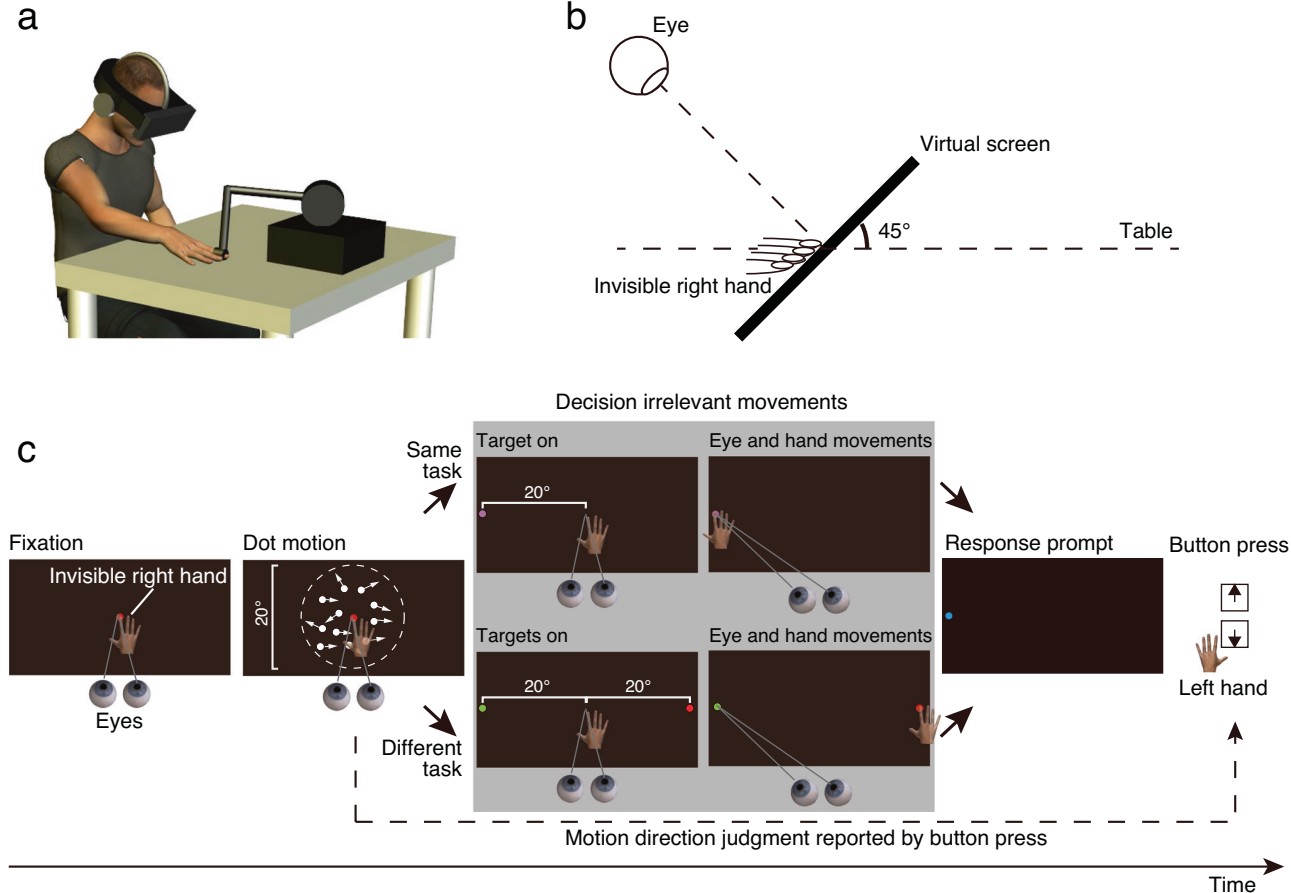

**Fig. 1 Measuring saccade and reach metrics during an ongoing perceptual judgement task. a** Participants binocularly viewed the visual stimulus presented through a head-mounted display and placed their right hand on a table. A position sensor was attached to the tip of the participant's right index finger. **b** Visual stimuli were presented on a black virtual screen inclined 45° with respect to the table in the virtual environment. The participant's eyes were initially directed to the centre of the virtual screen. The participant's unseen right index finger was initially placed in the centre of the virtual screen. **c** In a given trial, random-dot motion stimuli were presented after the participant fixated for 1 s. At the offset of the motion stimuli, saccade and reach targets appeared to either the left or right side of the virtual screen. Participants made simultaneous saccade and reach movements towards the targets as quickly as possible. After the two movements, they were instructed to report the motion direction by pressing a button with their left hand. The dashed lines depicting the aperture in the dot motion were not shown in the experiments. An example trial from experiment 1 is shown here.

pressing a button. Importantly, in between the motion stimulus and the judgement response, saccade and/or reach targets were presented following the offset of the motion stimulus. In experiment 1, participants had to make simultaneous saccade and reach movements towards these judgement-irrelevant targets. In experiment 2, participants had to make a single saccade or reach movement towards the judgement-irrelevant target. After these movements were made, the perceptual judgement was reported. By probing the oculomotor and manual systems with simultaneous judgement-irrelevant saccade and reach movements during judgement formation, we were able to investigate whether the oculomotor system was interfered with more greatly than the manual system during perceptual decision making (experiment 1). By probing the manual system with judgement-irrelevant reach movements without saccades during judgement formation, we were able to investigate whether the manual system was interfered with during perceptual decision making (experiment 2). Our results demonstrate that perceptual decision making only interferes with saccadic eye movements when simultaneous saccade and reach movements are made, suggesting that perceptual decision-related signals continuously flow into the oculomotor system alone even when multiple judgement-irrelevant actions are performed.

## Results

**Experiment 1: combined judgement-irrelevant saccade and reach movements.** We examined whether there are interactions between motion strength and saccade-reach generation in the context of a motion direction discrimination task. To this end, we used two conditions: (i) active decision making, in which participants actively discriminated the direction of visual motion, and (ii) passive viewing, in which the participants were not prompted to report the direction of motion. Comparison between these two conditions allowed us to test whether the effects on movement are caused by decision making or just by the viewing of visual motion. After participants viewed the visual motion, they performed a combined eye–hand movement task in which the axis of visual motion for direction discrimination was perpendicular to the axis of saccade and/or reach target locations (i.e., up-versus-down visual motion and left-or-right saccade and reach targets). This configuration would lead to a direction-specific shift in spatial attention either above or below the motion aperture, both of which were spatially unrelated to the left/right saccade or reach. Therefore, the direction of visual motion could not serve as an attentional cue to facilitate saccade and reach reaction times[30].

Participants placed their right palm down on a table and wore a head-mounted display (HMD) that displayed visual stimuli. In a

given trial, after participants fixated on the fixation point and pressed a button with their left hand to start the trial, a random-dot motion stimulus of variable coherence (3%, 6%, 12%, 24% or 48%) was presented for 100 ms around the fixation point. After the offset of the motion stimulus, saccade and reach targets were displayed on either the left or right side of the display, 20° horizontally from the fixation point. Participants were instructed to make simultaneous saccade and reach (right hand) movements towards either the same target (same task) or different targets (different task; one for saccade and one for reach) as quickly as possible. After the saccade and reach, they reported their motion direction judgement by pressing keyboard buttons with the fingers of their left hand. This stimulus sequence is illustrated in Fig. 1. The motion direction could be either up or down, and the saccade and reach targets could appear in either the left or right visual field. This made the simultaneous saccade and reach movements irrelevant to motion direction discrimination.

In the active decision-making condition, motion direction discrimination accuracy varied systematically with motion coherence, confirming that participants were engaged in the judgement task (Fig. 2a; see Supplementary Fig. 1 for individual data). The psychometric functions were almost identical between the same and different tasks [$F(1, 14) = 0.58$, $P = 0.46$, $\eta_p^2 = 0.54$], suggesting that congruency between saccade and reach directions did not affect psychophysical performance, even though saccade and reach took place in between motion viewing and the subsequent perceptual judgement response. This supports the view that participants treated the saccade and reach task as irrelevant to the performance of the perceptual judgement task. However, Bayes factor analysis provided inconclusive evidence for the null hypothesis that the psychometric functions were the same between the two tasks [$BF_{01}$ (Bayes factor) $= 0.96$].

We analysed saccade and reach reaction times and saccade and reach peak velocities to determine how perceptual judgements influence judgement-irrelevant saccade and reach movements. In the conventional paradigms used to study perceptual judgements, the reaction times and peak velocities of saccade or reach choices are mapped as functions of a sensory stimulus feature dimension[2,18,31,32]. Interestingly, studies using a motor-choice task have shown that saccade and reach reaction times and saccade peak velocities reflect different perceptual decision processes. While saccade and reach reaction times have been shown to reflect the expectation of the reward or value associated with the saccade target[33,34], there is also evidence that saccade peak velocities reflect the degree of certainty with which a perceptual decision is made (i.e., confidence in a decision)[32,35,36]. Based on these findings, we focused on the reaction times and peak velocities of judgement-irrelevant saccade and reach movements.

Saccade reaction times to the judgement-irrelevant saccade target were systematically affected by motion coherence in the active decision-making condition (solid symbols in Fig. 2b, c; see Supplementary Fig. 2 for individual data). There were significant interactions between the decision-making condition and motion coherence level [same task, $F(4, 28) = 6.90$, $P = 0.00054$, $\eta_p^2 = 0.50$; different task, $F(4, 28) = 2.73$, $P = 0.049$, $\eta_p^2 = 0.28$]. Saccade reaction times were longer when participants made a judgement based on weaker motion strength and progressively shortened with increased motion coherence [same task, $F(4, 28) = 5.40$, $P = 0.0024$, $\eta_p^2 = 0.26$; different task, $F(4, 28) = 2.98$, $P = 0.036$, $\eta_p^2 = 0.15$]. In contrast, although the visual stimuli in the passive viewing condition were identical to those in the active decision-making condition, motion coherence no longer modulated saccade reaction times in the passive viewing condition

(open symbols in Fig. 2b, c) [same task, $F(4, 28) = 1.47$, $P = 0.24$, $\eta_p^2 = 0.17$; different task, $F(4, 28) = 0.43$, $P = 0.79$, $\eta_p^2 = 0.44$]. These results suggest that saccade reaction times are affected by perceptual decision making when simultaneous saccade and reach movements are made. However, for the same and different tasks, Bayes factor analysis provided inconclusive evidence for the null hypothesis that motion coherence did not modulate saccade reaction times in the passive viewing condition [same task, $BF_{01} \approx 0.55$; different task, $BF_{01} \approx 1.61$].

Reach reaction times to the judgement-irrelevant reach target were not affected by motion coherence in either the active decision-making condition or passive viewing condition (Fig. 2d, e; see Supplementary Fig. 2 for individual data). There were no significant main effects of decision-making condition (active and passive) [same task, $F(1, 7) = 0.30$, $P = 0.59$, $\eta_p^2 = 0.041$; different task, $F(1, 7) = 0.68$, $P = 0.41$, $\eta_p^2 = 0.088$], although Bayes factor analysis provided inconclusive evidence for the null hypothesis that reach reaction times were the same between the active and passive conditions for the same and different tasks [same task, $BF_{01} = 1.92$; different task, $BF_{01} = 1.41$]. There were no significant main effects of motion coherence level [same task, $F(4, 28) = 0.68$, $P = 0.61$, $BF = 0.071$, $\eta_p^2 = 0.088$; different task, $F(4, 28) = 0.70$, $P = 0.60$, $BF_{01} = 11.36$, $\eta_p^2 = 0.090$]. There were also no significant interactions between the decision-making condition and motion coherence level [same task, $F(4, 28) = 0.44$, $P = 0.78$, $BF_{01} = 7.69$, $\eta_p^2 = 0.059$; different task, $F(4, 28) = 0.19$, $P = 0.94$, $BF_{01} = 8.33$, $\eta_p^2 = 0.027$]. These results suggest that reach reaction times are not affected by perceptual decision making when individuals are making simultaneous saccade and reach movements.

Saccade peak velocities were not affected by motion coherence in either the active decision-making or passive viewing conditions. Saccade peak velocities for the same and different tasks as a function of motion coherence are depicted in Fig. 3a, b, respectively (see Supplementary Fig. 3a–d for individual data). There were no significant main effects of decision-making condition (active and passive) [same task, $F(1, 7) = 0.0$, $P = 0.99$, $BF_{01} = 4.35$, $\eta_p^2 = 0.0006$; different task, $F(1, 7) = 0.40$, $P = 0.55$, $BF_{01} = 2.38$, $\eta_p^2 = 0.054$], although Bayes factor analysis provided inconclusive evidence for the null hypothesis that there were no main effects of decision-making condition for the different task. There were no significant main effects of motion coherence level [same task, $F(4, 28) = 0.78$, $P = 0.55$, $BF_{01} = 10.75$, $\eta_p^2 = 0.10$; different task, $F(4, 28) = 1.21$, $P = 0.33$, $BF_{01} = 8.33$, $\eta_p^2 = 0.15$]. There were also no significant interactions between the decision-making condition and motion coherence level [same task, $F(4, 28) = 1.64$, $P = 0.19$, $BF_{01} = 1.39$, $\eta_p^2 = 0.19$; different task, $F(4, 28) = 0.23$, $P = 0.92$, $BF_{01} = 7.69$, $\eta_p^2 = 0.031$], although Bayes factor analysis provided inconclusive evidence for the null hypothesis that there were no interactions between the decision-making condition and motion coherence level for the same task. Unlike saccade reaction times, these results suggest that motion coherence does not affect the saccade peak velocity in the active decision-making condition or in the passive viewing condition.

Reach peak velocities were also not affected by motion coherence in either the active decision-making or passive viewing conditions. Reach peak velocities for the same and different tasks as a function of motion coherence are depicted in Fig. 3c and d, respectively (see Supplementary Fig. 3e–h for individual data). For the same task, there was no significant main effect of decision-making condition (active and passive) [$F(1, 7) = 3.39$,

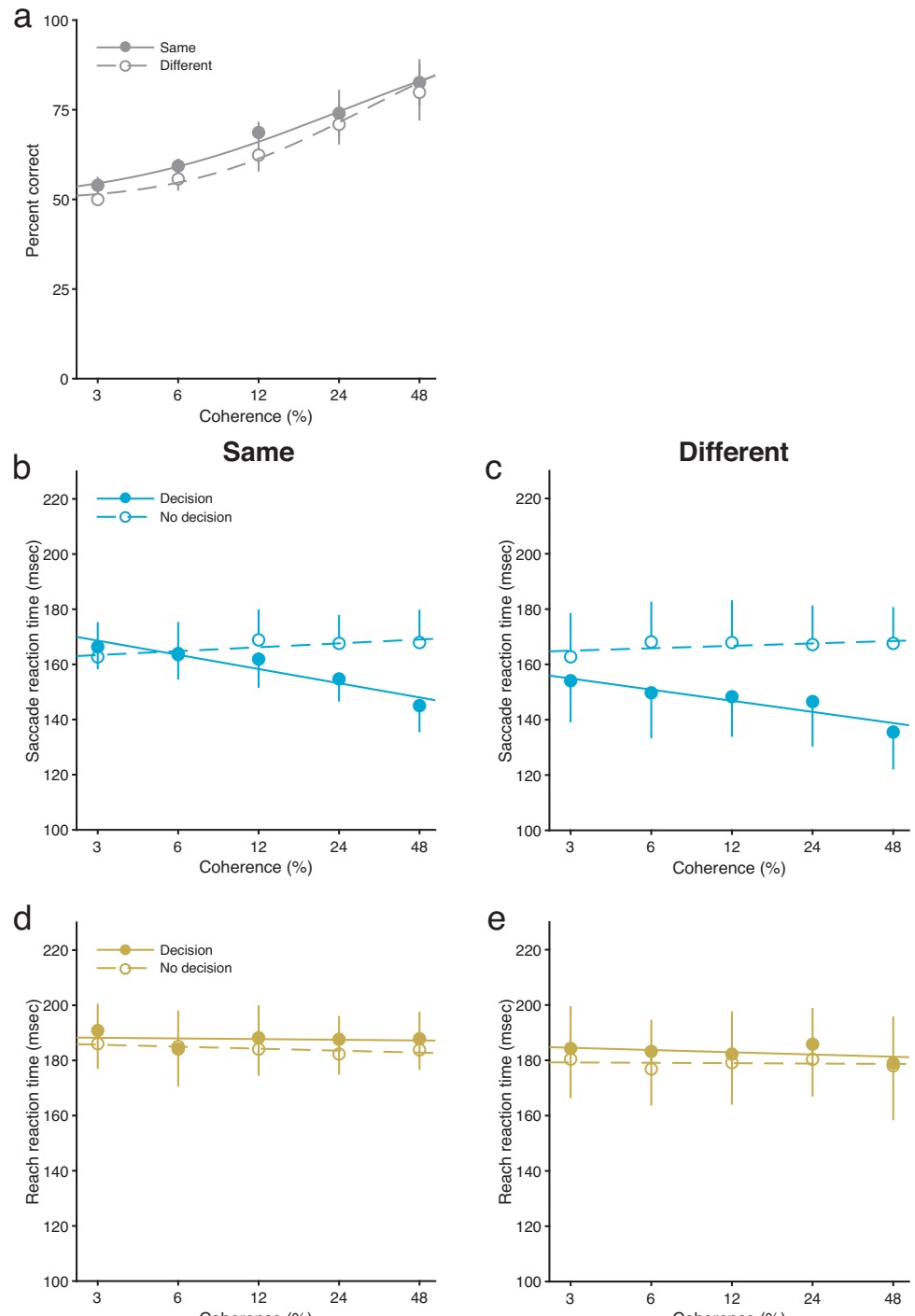

**Fig. 2 Motion direction discrimination accuracy, and the reaction times of simultaneous judgement-irrelevant saccade and reach movements.**
**a** Psychometric functions for the same and different tasks as a function of motion coherence. Saccade reaction times for the same (**b**) and different (**c**) tasks as a function of motion coherence. Reach reaction times for the same (**d**) and different (**e**) tasks as a function of motion coherence. Solid and dashed lines are the fitted lines for the active decision-making and passive viewing conditions, respectively. Results are the mean ± standard error. Error bars represent standard errors. $n = 8$.

$P = 0.11$, $\eta_p^2 = 0.33$], although Bayes factor analysis showed that reach peak velocities were substantially smaller in the active condition than in the passive condition [$BF_{10} = 3714.28$] (see Supplementary Fig. 4 for the dependence of the $BF_{10}$ on prior width). For the different task, there was no significant main effect of decision-making condition (active and passive) [$F(1, 7) = 0.51$, $P = 0.50$, $\eta_p^2 = 0.067$], although Bayes factor analysis provided

inconclusive evidence for the null hypothesis that the main effect of decision-making condition was not valid [$BF_{01} = 1.52$]. There were no significant main effects of motion coherence level [same task, $F(4, 28) = 2.01$, $P = 0.12$, $BF_{01} = 10.31$, $\eta_p^2 = 0.22$; different task, $F(4, 28) = 1.23$, $P = 0.32$, $BF_{01} = 10.0$, $\eta_p^2 = 0.15$]. There were also no significant interactions between the decision-making condition and motion coherence level [same task, $F(4, 28) = 1.02$,

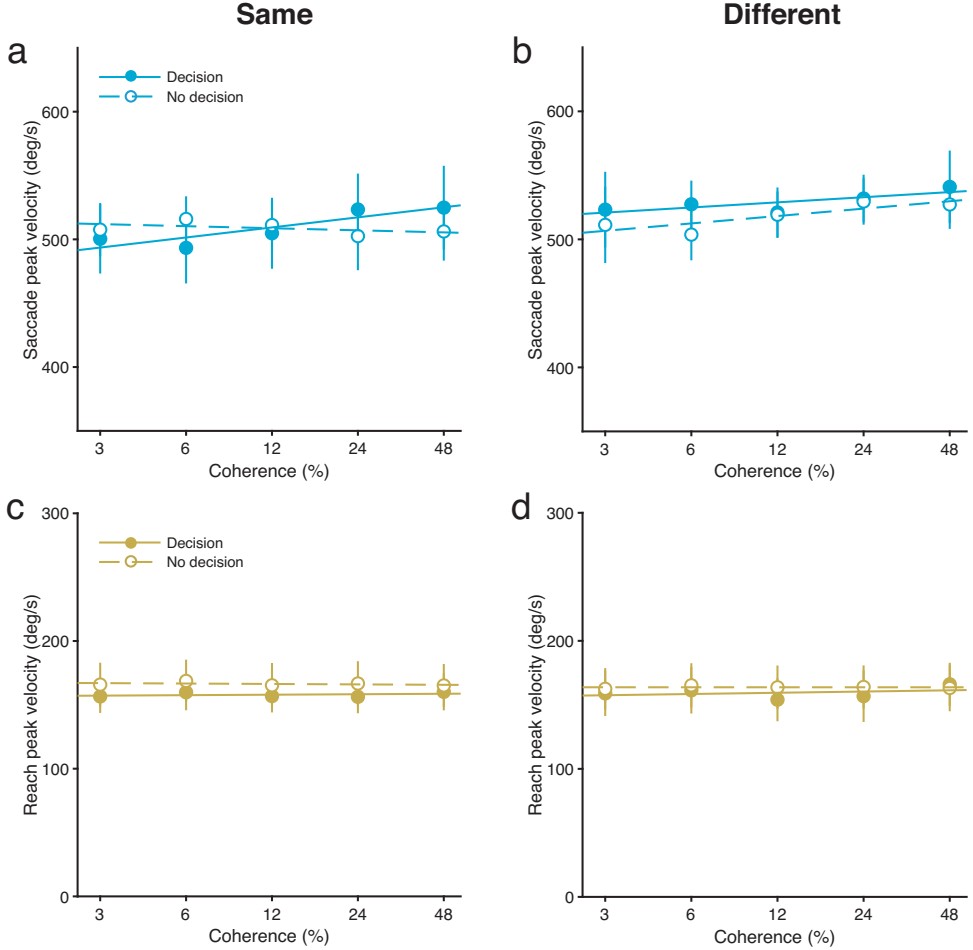

**Fig. 3 Peak velocities of simultaneous judgement-irrelevant saccade and reach movements.** Saccade peak velocities for the same (**a**) and different (**b**) tasks as a function of motion coherence. Reach peak velocities for the same (**c**) and different (**d**) tasks as a function of motion coherence. Solid and dashed lines are the fitted lines for the active decision-making and passive viewing conditions, respectively. Results are the mean ± standard error. Error bars represent standard errors. $n = 8$.

$P = 0.41$, $BF_{01} = 7.14$, $\eta_p^2 = 0.13$; different task, $F(4, 28) = 0.92$, $P = 0.47$, $BF_{01} = 5.0$, $\eta_p^2 = 0.12$]. Similar to reach reaction times, these results suggest that motion coherence does not affect the reach peak velocity in the active decision-making condition or in the passive viewing condition.

Overall, our results show that saccade reaction times are modulated by motion strength when individuals make simultaneous saccade and reach movements. However, this was not the case for reach reaction time, saccade peak velocity and reach peak velocity.

It may be a concern that the lack of alteration in the reach reaction time relative to the saccade reaction time reflected the fact that reach reaction times were much longer than saccade reaction times. We analysed the actual reaction times to examine whether reach reaction times are much longer than saccade reaction times. Reaction times for all conditions (i.e., viewing condition in the motion direction discrimination task, motion coherence and movement direction) were averaged for both saccade and reach movements. Reach reaction times were not significantly longer than saccade reaction times [Fig. 4a; $F(1, 7) = 3.47$, $P = 0.10$, $\eta_p^2 = 0.33$], although the difference in reaction times between saccade and reach was 23.95 ms in experiment 1, as shown in Fig. 4a. This suggests that reach reaction times were not much longer than saccade reaction times. In addition, the standard deviation of reaction times was not

significantly different between saccade and reach [Fig. 4b; $F(1, 7) = 0.02$, $P = 0.89$, $\eta_p^2 = 0.0028$], suggesting that the lack of alteration in reach reaction times with motion coherence cannot be explained by differences in reaction time variability. Thus, these results rule out the possibility that the lack of alteration in reach reaction time relative to saccade reaction time could reflect the fact that reach reaction times are much longer than saccade reaction times. However, Bayes factor analysis provided inconclusive evidence for two null hypotheses: (i) reach reaction times were not longer than saccade reaction times [$BF_{01} = 0.79$], and (ii) the standard deviation of reaction times was not different between saccade and reach [$BF_{01} = 2.33$].

**Experiment 2: single judgement-irrelevant saccade or reach movement.** To test whether simultaneous saccade and reach movements eliminate the modulation of reach reaction times by motion strength, we conducted a single-movement task experiment in which participants either made a saccade to the target without reaching towards it (saccade-only task) or a reach towards the target without looking at it (reach-only task) under the active decision-making condition. If simultaneous saccade and reach movements are necessary to eliminate the modulation of reach reaction times by motion strength, this modulation should appear in the reach-only task. In addition, the modulation

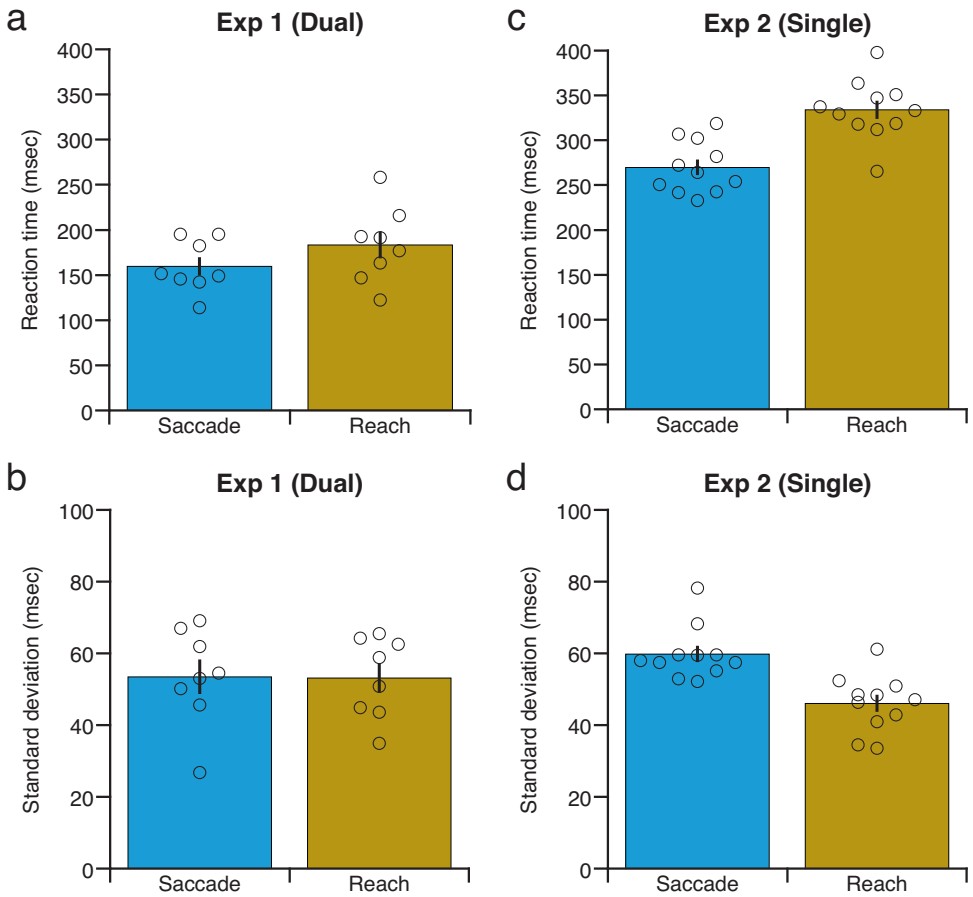

**Fig. 4 Analyses of actual reaction times to examine whether reach reaction times are much longer than saccade reaction times. a**, **c** Reaction times. **b**, **d** Standard deviations. Circle symbols represent different participants. Results are the mean ± standard error. Error bars represent standard errors. $n = 8$ for experiment 1. $n = 11$ for experiment 2.

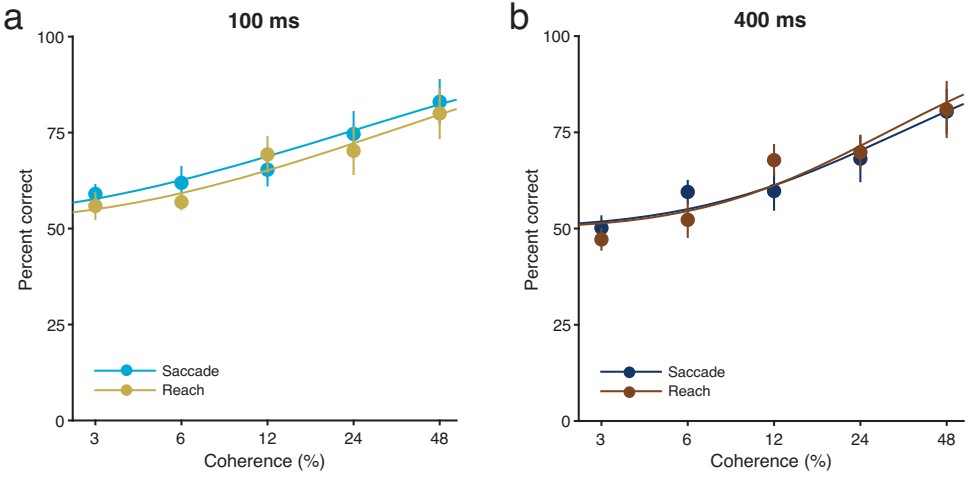

**Fig. 5 Motion direction discrimination accuracy for the short (100 ms) and long (400 ms) duration conditions as a function of motion coherence in the saccade-only and reach-only tasks. a** Short (100 ms) duration condition. Bright blue and yellow lines are the fitted lines for the saccade and reach movement conditions, respectively. **b** Long (400 ms) duration condition. Dark blue and brown lines are the fitted lines for the saccade and reach movement conditions, respectively. Results are the mean ± standard error. Error bars represent standard errors. $n = 11$.

of saccade reaction times by perceptual decision making should reappear in the saccade-only task.

In experiment 2, a random-dot motion stimulus of variable coherence (3%, 6%, 12%, 24% or 48%) and variable duration (100 or 400 ms) was presented around the fixation point. As in experiment 1, motion direction discrimination accuracy varied systematically with motion coherence in both the saccade-only and reach-only tasks (Fig. 5a, b; see Supplementary Fig. 5 for individual data). These results confirm that participants were engaged in making a perceptual judgement in both the

saccade-only and reach-only tasks. The psychometric functions were nearly identical in the saccade-only and reach-only tasks [$F(1, 10) = 1.77$, $P = 0.21$, $\eta_p^2 = 0.15$ for 100-ms duration; $F(1, 10) = 0.00$, $P = 0.99$, $\eta_p^2 = 0.00$ for 400-ms duration], indicating that psychophysical performance was not affected by the difference in the tasks, even though the saccade or reach took place in between motion viewing and the subsequent perceptual decision response. This supports the view that participants treated the saccade or reach task as irrelevant to the performance of the perceptual decision task. However, for 100-ms but not 400-ms duration, Bayes factor analysis provided inconclusive evidence for the null hypothesis that the psychometric functions were identical in the saccade-only and reach-only tasks [$BF_{01} = 2.94$ for 100-ms duration; $BF_{01} = 5.0$ for 400-ms duration].

As expected, motion coherence modulated not only saccade reaction times (Fig. 6a, c; saccade-only task; see Supplementary Fig. 6a, c for individual data) [$F(4, 40) = 3.72$, $P = 0.011$, $\eta_p^2 = 0.27$], but also reach reaction times (Fig. 6b, d; reach-only task; see Supplementary Fig. 6b, d for individual data) [$F(4, 40) = 5.98$, $P = 0.00072$, $\eta_p^2 = 0.37$]. There were significant differences between the two durations (100 and 400 ms) for both saccade reaction times [$F(1, 10) = 188.11$, $P = 8.24 \times 10^{-8}$, $\eta_p^2 = 0.95$] and reach reaction times [$F(1, 10) = 64.75$, $P = 1.12 \times 10^{-5}$, $\eta_p^2 = 0.87$]. However, there were no significant interactions between duration and motion coherence [saccade, $F(4, 40) = 1.52$, $P = 0.21$, $BF_{01} = 9.09$, $\eta_p^2 = 0.13$; reach, $F(4, 40) = 0.81$, $P = 0.53$, $BF_{01} = 10.53$, $\eta_p^2 = 0.075$]. Thus, judgement-irrelevant reach reaction times were modulated by motion coherence when individuals reached towards the target without looking at it. This indicates an interaction between perceptual judgements and the initiation of reach movements. Given that this interaction occurs when individuals make a reach without a saccade, the finding that judgement-irrelevant reach reaction times are not modulated by motion coherence when simultaneous saccade and reach movements are made (Fig. 2d, e) suggests that simultaneous saccade and reach movements are involved in preventing an interaction between perceptual decision making and the initiation of reach movements.

To examine the difference in reaction times between saccade and reach movements, we compared saccade reaction times in the saccade-only task and reach reaction times in the reach-only task. Reaction times for all conditions (i.e., motion coherence and duration) were averaged for both saccade and reach movements. Reach reaction times were significantly longer than saccade reaction times [Fig. 4c; $F(1, 10) = 45.97$, $P = 4.86 \times 10^{-5}$, $\eta_p^2 = 0.82$]. The standard deviation of reaction times was also significantly different between saccade and reach [Fig. 4d; $F(1, 10) = 31.26$, $P = 0.00023$, $\eta_p^2 = 0.76$]. Nevertheless, not only saccade reaction times, but also reach reaction times were modulated by motion coherence in experiment 2.

A possible concern is that the lack of alteration in the reach reaction time in the dual task (experiment 1) reflected the fact that reach reaction times were much longer in the dual task (experiment 1) than in the single task (experiment 2). We analysed the difference in reaction times between the dual and single tasks (Fig. 4a, c). A two-way analysis of variance (ANOVA) was performed with effectors and tasks as factors. There was a significant interaction between effector and task [$F(1, 17) = 6.65$, $P = 0.020$, $\eta_p^2 = 0.28$]. Reach reaction times were significantly shorter in the dual task than in the single task [$F(1, 17) = 76.11$, $P = 1.10 \times 10^{-7}$, $\eta_p^2 = 0.82$]. The same was true for saccades [$F(1, 17) = 66.93$, $P = 2.69 \times 10^{-7}$, $\eta_p^2 = 0.80$]. Thus, these results rule

out the possibility that the lack of alteration in the reach reaction time in the dual task could reflect the fact that reach reactions times are much longer in the dual task than in the single task.

Another concern may be that the lack of alteration in the reach reaction time in the dual task (experiment 1) reflected the fact that the variance of reach reaction times was larger in the dual task (experiment 1) than in the single task (experiment 2). We analysed the difference in the mean standard deviation of reaction times between the dual and single tasks (Fig. 4b, d). A two-way ANOVA was performed with effectors and tasks as factors. There was a significant interaction between effector and task [$F(1, 17) = 14.60$, $P = 0.0014$, $\eta_p^2 = 0.46$]. The mean standard deviation of reach reaction times was not significantly different between the dual and single tasks [$F(1, 17) = 2.61$, $P = 0.12$, $\eta_p^2 = 0.13$]. Moreover, the mean standard deviation of saccade reaction times was not significantly different between the dual and single tasks [$F(1, 17) = 1.72$, $P = 0.21$, $\eta_p^2 = 0.092$]. Thus, these results rule out the possibility that the lack of alteration in the reach reaction time in the dual task reflected the fact that the variance of reach reaction times was larger in the dual task than in the single task. However, for reaches and saccades, Bayes factor analysis provided inconclusive evidence for the null hypothesis that the mean standard deviation of reaction times was not different between the dual and single tasks [reach, $BF_{01} \approx 1.03$; saccade, $BF_{01} \approx 1.37$].

To examine whether the modulation of saccade or reach peak velocities by motion strength is also eliminated by simultaneous saccade and reach movements, we analysed saccade peak velocities in the saccade-only task and reach peak velocities in the reach-only task. We expected that saccade peak velocities would be modulated by motion strength in the saccade-only task, because a previous study has reported such a modulation[18]. Indeed, motion coherence did modulate saccade peak velocities in the saccade-only task (Fig. 6e; see Supplementary Fig. 7a, c for individual data) [$F(4, 40) = 2.56$, $P = 0.045$, $\eta_p^2 = 0.18$]. There was no significant difference in saccade peak velocity between the two durations (100 and 400 ms) [$F(1, 10) = 0.06$, $P = 0.81$, $BF_{01} = 3.85$, $\eta_p^2 = 0.024$]. There was also no significant interaction between duration condition and motion coherence level [$F(4, 40) = 0.32$, $P = 0.86$, $BF_{01} = 10.0$, $\eta_p^2 = 0.037$]. Thus, judgement-irrelevant saccade peak velocities were modulated by motion coherence when individuals made a saccade to a target without reaching towards it. This indicates that perceptual judgements modulate not only saccade reaction times, but also saccade peak velocities. Given that this modulation of saccade peak velocities occurs only when individuals make a saccade without a reach, the finding that judgement-irrelevant saccade peak velocities are not modulated by perceptual judgements when simultaneous saccade and reach movements are made (Fig. 3a, b) suggests that simultaneous saccade and reach movements are also involved in preventing an interaction between perceptual decision making and the velocity of saccade movements.

Surprisingly, motion coherence did not modulate reach peak velocities, even in the reach-only task (Fig. 6f; see Supplementary Fig. 7b, d for individual data) [$F(4, 40) = 2.03$, $P = 0.11$, $BF_{01} = 3.23$, $\eta_p^2 = 0.17$]. There was no significant difference between the two durations (100 and 400 ms) for reach peak velocities [$F(1, 10) = 4.11$, $P = 0.070$, $\eta_p^2 = 0.29$], but Bayes factor analysis showed that reach peak velocities was substantially larger in the 100-ms duration condition than in the 400-ms duration condition [$BF_{10} = 47.04$]. There was no significant interaction between duration condition and motion coherence level [$F(4, 40) = 0.60$, $P = 0.66$, $BF_{01} = 8.33$, $\eta_p^2 = 0.056$]. These results indicate that perceptual judgements do not modulate reach peak

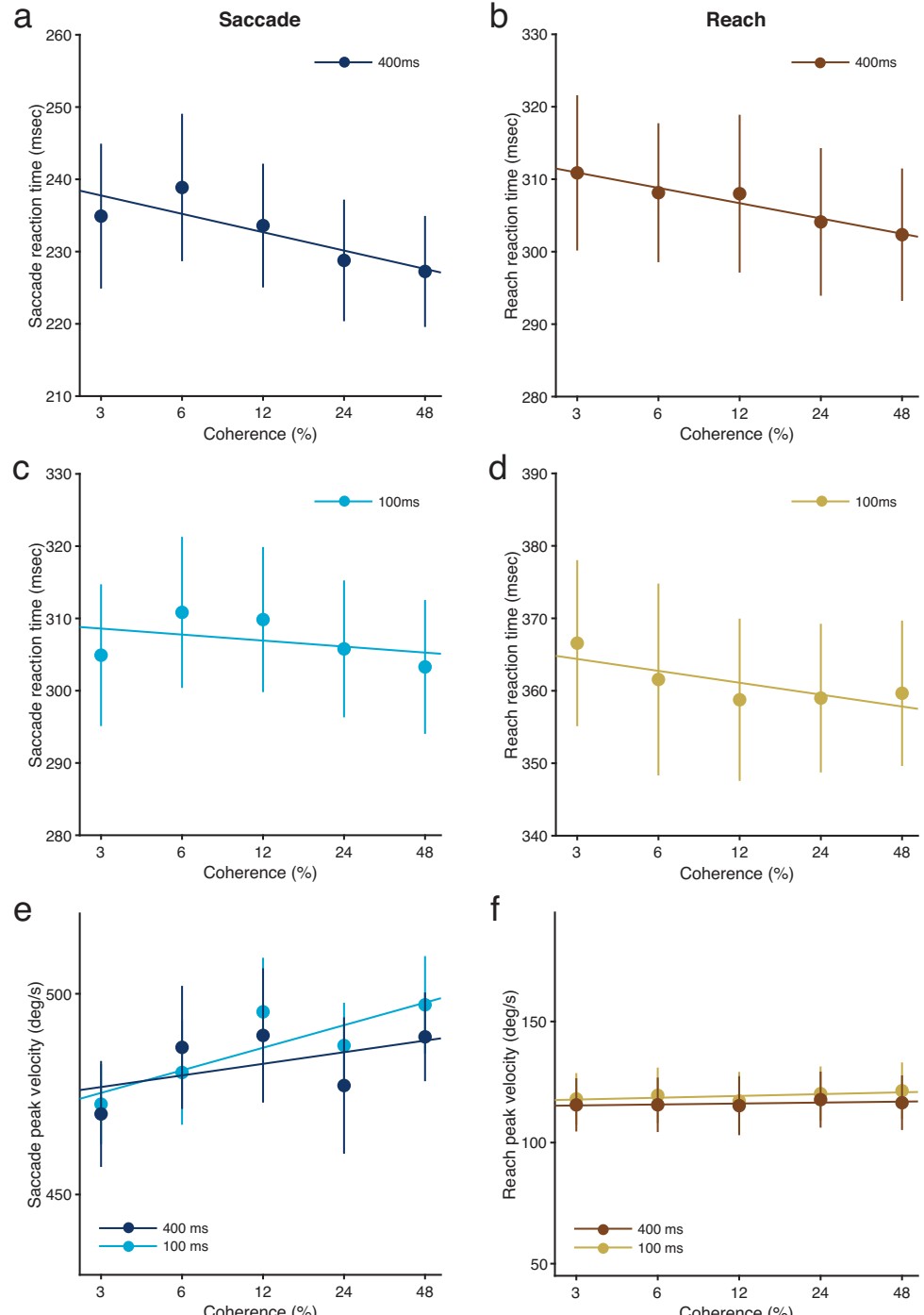

**Fig. 6 Reaction times and peak velocities for saccades and reaches as a function of motion coherence in the saccade-only and reach-only tasks.**
Reaction times for the saccade (**a**, **c**) and reach (**b**, **d**) movement conditions as a function of motion coherence. **a**, **b** The long (400 ms) duration condition. **c**, **d** The short (100 ms) duration condition. Individual data for **a**–**d** are shown in Supplementary Fig. 6. Peak velocities for the saccade (**e**) and reach (**f**) movement conditions as a function of motion coherence. Individual data for **e**, **f** are shown in Supplementary Fig. 7. Results are the mean ± standard error. Error bars represent standard errors. $n = 11$.

velocities, regardless of whether a reach is made with or without a saccade. This suggests that reach peak velocities do not reflect perceptual decision-making processes when judgement-irrelevant reach movements are made.

To summarise experiment 2, reach reaction times and saccade peak velocities were modulated by motion coherence in the reach-only task and saccade-only task, respectively. In contrast, reach peak velocities were not modulated by motion coherence, even in

the reach-only task. These results suggest that simultaneous saccade and reach movements are required to eliminate the modulations of reach reaction times and saccade peak velocities by motion strength.

## Discussion
Neurophysiological studies have shown that perceptual judgement-related activity arises in both oculomotor and manual

brain areas such as the LIP[1,12,17,20] and MIP[17], respectively. These brain areas show selectivity for both the formation of perceptual judgements and the execution of movements within the same neurons. This leads to the view that perceptual judgements interfere with judgement-irrelevant movements. Indeed, perceptual judgement-related signals have been observed to interfere with judgement-irrelevant saccade responses in the LIP of monkeys[12–16]. This has been supported by a human behavioural study in which perceptual judgements interfered with judgement-irrelevant eye movements[18]. Interestingly, a recent neurophysiological study has shown that the activity of MIP neurons is greatly attenuated when perceptual judgements are communicated by eye movements while LIP neurons still activate when perceptual judgements are communicated by reach movements, as well as eye movements[17]. We investigated whether perceptual judgements interfere only with eye movements when simultaneous judgement-irrelevant eye and reach movements are made. Furthermore, given that the MIP shows selectivity for both the formation of perceptual judgements and the execution of reach movements within the same neurons[17], we expected that perceptual judgement-related activity might interfere with judgement-irrelevant reach movements when reach movements are made without eye movements. Through these investigations, the present study reveals that perceptual judgements interfere more with eye movements than with reach movements when simultaneous judgement-irrelevant eye and reach movements are made and that perceptual judgements interfere with judgement-irrelevant reach movements when reaches are made without saccades.

We found that saccadic eye movements were only affected by the motion strength that informed the perceptual decisions about the direction of the visual motion when simultaneous saccade and reach movements were made towards targets that were irrelevant to the motion discrimination task (experiment 1: the dual-movement task). Specifically, saccade, but not reach, reaction times were affected in proportion to motion strength. This finding appeared only with simultaneous saccade and reach movements during an active decision-making task. Passive viewing of a motion stimulus did not have the same effects on saccade reaction times. The results of the dual-movement task indicate that perceptual judgements about visual motion interfere only with saccade reaction times when simultaneous judgement-irrelevant saccade and reach movements are made. In addition, when reach movements were made to a judgement-irrelevant target without saccades, reach reaction times were modulated by the motion strength (experiment 2: the single-movement task). The results of the single-movement task indicate that perceptual judgements about visual motion can interfere with judgement-irrelevant reach reaction times. These results suggest that perceptual judgements about visual motion interfere more with saccade reaction times than with reach reaction times when simultaneous judgement-irrelevant saccade and reach movements are made.

We also found that perceptual judgements did not affect saccade peak velocities when simultaneous judgement-irrelevant saccade and reach movements were made. No modulation of saccade peak velocities by motion strength was observed in the dual-movement task (Fig. 3a, b). However, saccade peak velocities were modulated by motion strength in the single-movement task (Fig. 6e). These results indicate that, although perceptual decision-making processes can interfere with the oculomotor system that influences saccade peak velocity, this interference disappears when simultaneous saccade and reach movements are made. This suggests that simultaneous saccade and reach movements are involved in preventing the interference between perceptual decision making and saccade velocity.

In contrast, we found that perceptual judgements did not affect reach peak velocities, regardless of whether reaches were made with or without saccades. No modulation of reach peak velocities by motion strength was observed in both dual- and single-movement tasks (Fig. 3c, d for the dual task; Fig. 6f for the single task). These results suggest that perceptual decision-making processes themselves do not interfere with the manual system that influences reach peak velocity.

Our results suggest that perceptual decision making, rather than spatial attention, modulates judgement-irrelevant saccade and reach movements. Strong rightward visual motion has been suggested to serve as an attentional cue to the right, and vice versa[30,37,38]. For example, an attentional cue to the right served by rightward visual motion would facilitate a rightward saccade or reach reaction time. However, in the present study, the axis of visual motion for direction discrimination was perpendicular to the axis of the motor target locations (i.e., up versus down visual motion, and left or right motor targets). In this geometric configuration, a direction-specific shift in spatial attention is either above or below the motion stimulus. Thus, an attentional shift would not be expected to modulate saccade or reach movements in our experiments, suggesting that spatial attention could not account for the present results.

Our results showed that there were individual variabilities in motion direction discrimination accuracy (Supplementary Figs. 1 and 5). Several participants had near chance performance even at the highest motion coherence. These participants may have prioritized the movement task over the perceptual decision task, perhaps because they found it difficult to move their eye and/or hand quickly in the movement task. Given that a low accuracy of motion direction discrimination judgements reflects perceptual decisions driven by weaker sensory evidence[39], it is possible that these participants may have a smaller influence of motion direction discrimination judgements on judgement-irrelevant saccade and reach movements. To test this possibility, we analysed how motion direction discrimination accuracy affects saccade reaction times, reach reaction times and saccade peak velocities. In this analysis, the degree of modulation of reaction time and velocity by motion coherence (we refer to this degree as the modulation index) was calculated as the slope of reaction time and velocity against motion coherence, respectively (see the caption of Supplementary Fig. 8 for more details). The modulation indices were classified into high-accuracy and low-accuracy groups. The high-accuracy group consisted of participants with 75% or more perceptual accuracy at the highest motion coherence level. We found that participants in the low-accuracy group had a significantly lower modulation index than participants in the high-accuracy group for saccade reaction times, reach reaction times and saccade peak velocities (Supplementary Fig. 8; $t_{16} = 3.60$, $P = 0.0012$, $d = 1.68$ for saccade reaction times; $t_6 = 2.45$, $P = 0.025$, $d = 1.42$ for reach reaction times; $t_7 = 2.35$, $P = 0.025$, $d = 1.60$ for saccade peak velocities). These results are consistent with an explanation that active perceptual decision-making processes affect judgement-irrelevant motor actions. However, because the individual differences in motion direction discrimination accuracy were enormous, it would be better to conduct an experiment with adjustment of the motion coherence level for each participant. Future research is needed to examine this.

Overall, our results show the following: (i) perceptual decision-making processes interfere more with saccade reaction times than with reach reaction times when simultaneous judgement-irrelevant saccade and reach movements are made; (ii) perceptual decision-making processes interfere with reach reaction times when judgement-irrelevant reach movements are made without saccades; (iii) perceptual decision-making processes do not

interfere with saccade peak velocities when simultaneous judgement-irrelevant saccade and reach movements are made; and (iv) perceptual decision-making processes do not interfere with reach peak velocities regardless of whether reach movements are made with or without saccades. These findings suggest that perceptual decision-related signals flow between the oculomotor and manual systems in a complicated way.

These complicated results may be explained based on the channel modulation hypothesis[19]. According to this hypothesis, the communication channel is formed from projections from the perceptual decision-making system to the motor system. As a result, activity in the perceptual decision-making system affects responses in the motor system. In addition, a modulator network is placed between the perceptual decision-making system and the motor system. The modulator network works through motor commands from eye and/or reach movements. Activity in the modulator network can alter the motor system response to input from the perceptual decision-making system, modulating the communication channel. Such a channel modulation could produce changes in reaction times and peak velocities for movements by opening the channel that communicates perceptual decision signals to guide a motor response (e.g., saccades) or by closing the channel that communicates perceptual decision signals to guide a different motor response (e.g., reaches). Thus, interactions between perceptual decision-making and motor responses in behaviour could be the result of channel modulations in multi-regional communication.

Our results indicate that simultaneous saccade and reach movements are required to eliminate the modulations of reach reaction times and saccade peak velocities, but not saccade reaction times, by motion strength. Saccade and reach reaction times have been reported to reflect the expectation of the reward or value associated with the movement target[33,34], and on the other hand saccade reaction times can also reflect the degree of certainty with which a perceptual decision is made (i.e., confidence in a decision)[40,41]. Meanwhile, saccade peak velocities have been reported to reflect confidence in a decision[32], and on the other hand an influence on saccade peak velocities from the reward or value associated with the saccade target has been demonstrated in many studies[42–44]. Although it is unclear whether reaction time and peak velocity reflect different or similar features of perceptual decisions, our findings suggest that simultaneous saccade and reach movements may be involved in suppressing the channel related to decision confidence in the oculomotor circuitry and the channel related to the expectation of a reward or the value in the manual circuitry, and may interact only with the channel related to the expectation of a reward or the value in the oculomotor circuitry. Thus, the concept of communication channel modulation between the perceptual decision-making system and the motor system may be a key to understanding the mechanisms underlying the link between decision-related activity and decision-unrelated motor processes.

Our results imply that the initiation of saccades is affected only during perceptual decision making, even when the perceptual judgements are not communicated with saccade and reach movements. What are the neural mechanisms underlying this interference between the oculomotor system and the perceptual decision-making system? Saccade and reach movements are believed to be controlled by several brain areas, including the posterior parietal cortex[45]. The posterior parietal cortex has several distinguishable subregions. In these subregions, the LIP and the parietal reach region (PRR) encode signals related to saccade and reach movements, respectively. In addition to such movement-related signals, the LIP[1] and PRR[46] are involved in decision making for saccade and reach movements, respectively. Many studies have suggested interactions between signals in the

oculomotor system and the manual system[47–50]. Interestingly, recent neurophysiological studies have indicated that perceptual decisions about where to make coordinated saccade and reach movements involve interactions between the LIP and PRR[49,50]. The present study suggests that these interactions may occur even when the perceptual decisions do not involve saccade and reach movements, implying that the LIP may be recruited and the PRR suppressed when simultaneous saccade and reach movements are made during perceptual decision making, despite the fact that these movements are not tied to perceptual decisions.

In conclusion, the present study reveals a tight coupling between oculomotor action and perceptual decisions, even in the planning of judgement-irrelevant saccade and reach movements, which indicates strong interference between oculomotor action and perceptual decision making. Moreover, saccade reaction times, but not reach reaction times, are shown to be modulated by decision formation. These findings suggest that oculomotor brain circuits may be recruited during perceptual decision making, even when multiple judgement-irrelevant motor actions are performed. This implies that decision signals continuously flow into the oculomotor system alone during perceptual decision making, even when multiple motor actions are judgement-irrelevant. Saccades might be useful to make inferences about covert perceptual decisions, even when perceptual decision making does not involve any motor choices.

## Methods

**Participants**. A total of 8 participants (2 women, 6 men; mean age 22.6 [range 19–25] years) were recruited for experiment 1, and 11 participants (3 women, 8 men; mean age 23.5 [range 19–30] years) were recruited for experiment 2. Participants were recruited from a paid participant pool (Sona Systems, Ltd.) and received a gift card of 1000 yen per hour for their participation. This paid participant pool comprised individuals who wished to participate in research studies being conducted by Tohoku University faculty members and graduate students. Undergraduate and graduate students of Tohoku University were among those registered in the paid participant pool. All participants had normal or corrected-to-normal vision and provided written informed consent in accordance with the Code of Ethics of the World Medical Association (Declaration of Helsinki). The study was approved by the Ethics Committee of the Graduate School of Information Sciences, Tohoku University. All methods were carried out in accordance with relevant guidelines and regulations.

**Apparatus and stimuli**. Participants placed their right hand on a table and wore an HMD (Tobii Pro VR Integration based on the HTC Vive, comprising dual 3.6-inch microdisplays with 1080 × 1200 pixels per eye and a 110° diagonal field-of-view; Tobii Technology, Stockholm, Sweden) that displayed visual stimuli[51]. The simulated viewing distance was 50 cm in the virtual space. The HMD was equipped with an eye tracker that measured the gaze position of the participant's eyes at a sampling rate of 120 Hz. The HMD was covered with black tissue to occlude all surrounding visual input and was equipped with a customised forehead rest. The arm of a PHANToM force-feedback device (3D Systems, Cary, NC) was attached to the participant's right index finger. The force-feedback device was used to measure the position of the participant's right hand at a sampling rate of 90 Hz.

Participants viewed random-dot motion stimuli (200 dots) that were displayed in an invisible circular aperture (20° in diameter) centred on a fixation point (red, 0.5° in diameter, 18.0 cd/m²). The fixation point was spatially aligned with the real table, where participants placed the tip of their right index finger at the start of each trial. Dots were white (50.0 cd/m²), subtended 0.6° and moved at a speed of 5°/s on a black background. Each dot was

assigned a random lifetime from a uniform distribution between 0 and 150 ms. When the lifetime of a dot expired, the dot was randomly placed within the aperture and assigned a lifetime of 150 ms. The central 3° around the fixation point was black. Motion coherence was defined as the percentage of dots moving together in the same direction among dots moving in random directions. We used five coherence levels: 3%, 6%, 12%, 24% and 48%. The motion direction of the random-dot stimulus was up or down and was perpendicular to the axis of the saccade or reach (left or right). The viewing durations of motion stimuli were 100 ms in experiment 1 and 100 or 400 ms in experiment 2.

**Procedures.** Each session started with an eye movement calibration procedure in which the participant fixated on ten dots presented sequentially on horizontal and vertical centre lines of the display and pushed a button upon completion of each fixation. Before the start of each trial, participants fixated on the fixation point at the centre of the display and placed the tip of their right index finger at the fixation point. Participants initiated a trial by pressing a button using their left hand. After a 1000-ms delay, random-dot motion stimuli were presented. At the offset of the motion stimulus, the fixation point disappeared and one or two targets appeared 20° to the right or left of the display centre. In experiment 1, participants performed a combined movement task, making simultaneous eye and hand movements towards the same location or two different locations. For the same task, one target (purple, 0.7° in diameter, 18.0 cd/m$^2$) appeared. For the different task, a saccade target (green, 0.7° in diameter, 18.0 cd/m$^2$) and a reach target (red, 0.7° in diameter, 18.0 cd/m$^2$) appeared. In experiment 2, participants performed a single-movement task, making an eye or hand movement towards the target location. For the eye movement task, participants were instructed to maintain their right index finger on the fixation point. For the hand movement task, they were instructed to keep their gaze on the fixation point. The colour of the targets changed from the original colours to blue (0.7° in diameter, 6.4 cd/m$^2$) 1200 ms after the target appearance. This prompted participants to report the direction of the random-dot motion. Participants used ten computer keyboard keys to report the motion direction with a left-finger button press. Auditory feedback was presented for the motion discrimination task (correct vs. incorrect).

In experiment 1, the combined movement task was used. Because participants made simultaneous eye and hand movements towards the same target or different targets, there were two conditions in this experiment: (i) same, in which the eye movement direction and hand movement direction were the same, and (ii) different, in which the eye movement direction was the opposite of the hand movement direction. In addition to these conditions, two more conditions were defined: (i) active decision making, in which participants actively discriminated the motion direction of the random-dot stimulus, and (ii) passive viewing, in which the participants were not prompted to report the motion direction. In experiment 2, all aspects of the experiment remained the same as in experiment 1, except that participants performed a single-movement task (eye movement only or hand movement only) and the passive viewing condition was not conducted. Condition order was counterbalanced across participants.

Each participant performed four sessions. In experiment 1, each session comprised four blocks (2 same movement directions × 2 different movement directions) of the active decision-making condition and four blocks (2 same movement directions × 2 different movement directions) of the passive viewing condition. Within each block, 40 trials (5 coherences × 2 motion directions × 4 repetitions) were conducted in a random order. The eye and hand movement directions were fixed across trials in a block. In experiment 2, each session comprised eight blocks of the eye

movement condition and eight blocks of the hand movement condition. Within each block, 80 trials (5 coherences × 2 motion directions × 2 movement directions × 2 durations × 2 repetitions) were conducted in a random order. Participants performed a practice block before the experimental sessions, and this block was not included in the data analysis.

**Analysis of eye and hand movements.** Saccade reaction times were defined as the time at which eye velocity exceeded 30°/s from saccade target presentation. Reach reaction times were defined as the time at which hand velocity exceeded 10°/s from reach target presentation. The following trials were rejected (21.5%): those in which participants failed to make the movements of the eye or hand as instructed and those in which the first saccade or reach amplitude was within 5°. The participants' fixation was relatively stable during the presentation of the motion stimulus (Supplementary Fig. 9).

**Statistics and reproducibility.** A total of 19 participants were recruited in this study. This sample size is typical for human psychophysical experiments. For statistical evaluation, we used repeated-measures analysis of variance (ANOVA). For experiment 1, a repeated-measures ANOVA was performed with two viewing conditions (active and passive) and five motion coherence levels (3%, 6%, 12%, 24% and 48%) as factors. For experiment 2, a repeated-measures ANOVA was performed with two duration conditions (100 ms and 400 ms) and five motion coherence levels (3%, 6%, 12%, 24% and 48%) as factors. In Fig. 4, a repeated-measures ANOVA was performed with two effectors (saccade and reach) as factors for each of experiments 1 and 2. For Supplementary Fig. 7, an unpaired $t$-test was performed with two accuracy groups (high-accuracy group and low-accuracy group). For Supplementary Fig. 8, a paired $t$-test was performed with two motion directions (upward and downward). Effect sizes $\eta_p^2$ and $d$ were calculated based on Cohen's definition for ANOVAs and $t$-tests, respectively[52]. We evaluated motion discrimination performance by defining a psychometric function. For the psychometric function, cumulative Gaussian functions were fit to the mean data of each condition with least squares regression. For reaction times and peak velocities, straight lines were fit to the mean data of each condition with least squares regression. This is because previous studies using a motion discrimination task in which a saccade was made in the perceived motion direction have shown that saccade reaction times fit well to a linear model[18,53]. The data were processed in MATLAB R2021a (MathWorks Inc., Natick, MA) and analysed using SPSS ver 25 (IBM Corp., Armonk, NY).

Bayes factors (BFs) were calculated using an R package for BF analysis[54,55]. In this package, Cauchy distribution was used as a prior distribution. Except that the scale setting for the prior distribution was set to 0.5, various settings followed the defaults of the package. The maximum number of estimations by the Markov chain Monte Carlo method was 10,000. Following the conventional scales for interpreting BF, cases with a BF less than 3.2 were rated as providing inconclusive evidence of the null or alternative hypothesis[56,57]. $BF_{01}$ and $BF_{10}$ represent indications of null and alternative hypothesis dominance, respectively.

**Reporting summary.** Further information on research design is available in the Nature Portfolio Reporting Summary linked to this article.

## Data availability
The data that support the findings of this study are available in an Open Science Framework repository (https://osf.io/vq95c/)[58].

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

## Acknowledgements

This work was partially supported by Japan Society for the Promotion of Science Grants-in-Aid for Scientific Research (JP22H00087 and JP21K12102 to K.M.).

## Author contributions

K.M. and S.F. designed the study; S.F. performed the experiments; S.F. and K.M. analysed the data; and K.M. wrote the manuscript.

## Competing interests

The authors declare no competing interests.
