## [Peer Review File · Communications Biology]

Referee expertise:

Referees #1-3: each have expertise in visual perception & saccades

Reviewers' comments:

Reviewer #1 (Remarks to the Author):

In this study, the authors asked human subjects to judge the direction of motion after performing an intermediate reach/saccade movement between stimulus presentations and response reporting. They found that reaction times and peak velocities of the intervening movements could be influenced by the stimulus under some circumstances.

The results are generally clear. One missing analysis, in my opinion, is the actual reaction times in the different paradigms. For example, the lack of alteration in reach reaction time relative to saccade reaction time could reflect the fact that reach reactions times are much longer than saccade reaction times.

Another comment is on improving clarity, especially in the early parts of the paper, as I list in the detailed comments below:

- line 43 (and abstract): I think it would be helpful in both the abstract and also early introduction to define and clarify things a bit more. In particular, I would recommend clarifying what is exactly meant by terms like "decision-irrelevant" and "interact". For example, in lines 17-18 of the abstract, I'm not sure what the difference is between "motor actions must be related to decision formation" and "perceptual decisions also influence a motor action unrelated to decisions"? Isn't the first already implying the second? I know that you probably want to say something else, but it needs to be crystal clear very early on in the paper. The introduction does a slightly better job of first saying that sometimes, one might think that motor actions are just an output stage, but in reality there might be reach curvatures etc that reflect the internal processing. But, again, the formulation of the entire thing needs to be much more clear. Returning back to line 43, I again see a statement like "eye movements that are decision irrelevant", but I still do not understand what is meant. I am sure it will get clearer later in the paper, but why leave it a mystery? Clarify as early as possible.

- line 45: I guess one problem I have is that the actions themselves involve decisions. So, when you say decision-irrelevant actions, I get confused. I think it is best to avoid this word decision as much as possible, and just say that subjects have to do a dual task: one perceptual judgement (or whatever else it is) and the other is a reach or eye movement. And, we show that the perceptual judgement interferes with the motor action even if the motor action is irrelevant to the judgement.

- line 56: what does it mean for the perceptual decision to prioritize decision-irrelevant eye movements? What does "prioritise" mean???? I think the message is that eye movements experience larger interference from perceptual decisions than reach movements. This is much more descriptive and appropriate in my opinion.

- line 61: add one sentence clarifying after the first sentence what you looked for. e.g. "We then looked for perturbations in reach or eye movement trajectory..." (or whatever else you measured...at this stage of the introduction, I still do not know what you measured yet).

- lines 62-64: now, suddenly the text says "eye and reach movements *affect* concurrent perceptual processes". Isn't it the perceptual process that interferes with the movement, at least up to this point in the abstract and introduction? I think the problem is that there is too much left to the reader in the text. It wouldn't hurt to explain paradigms and experiments rather than just list citations. This way,

confusions like mine at this sentence would be minimised.

- lines 66-75 are very nice

- line 120: can you assess this similarity of psychometric curves statistically? I think that this is critical to show.

- line 134: you might want to also look at Buonocore et al, 2017, 2021 providing a mechanistic explanation for why kinematics might be altered with sensory interference in the citation that you made

- line 157: One big difference between reaches and saccades is that the reaches probably have 3x the reaction times of the saccades. So, the lack of effect could simply be because there was too much time between the motion patch and the reach. You do not show the actual manual and saccadic reaction times, but I think you should show them. And, you should consider the possibility that the large difference in reaction time between saccades and reaches could explain your results.

- line 160: I think I expected this result (see comment above about the kinematic alterations)

- line 195: again, I cannot see whether they are "nearly identical" because they are put in different panels. Some statistical quantification and/or plotting in the same panel is necessary

- line 209: but maybe reach reaction time in the dual task was longer than in the single task. i.e. when required to both reach and look, the reaction times are longer than with only reaching. Similarly for saccades.

Reviewer #2 (Remarks to the Author):

This paper reports a set of puzzling findings about the interaction between perceptual decisions and the concurrent planning and execution of eye and hand movements that are irrelevant to the decision itself. Previous research (most notably Joo, Katz & Huk 2016) reported that perceptual decisions influenced decision-irrelevant eye movements, concluding it was evidence of motor and decision signals being mixed (multiplexed) in the same neural circuitry. (More specifically, the influence consisted in a modulation of the latency and peak velocity of decision-irrelevant saccadic eye movements.) In the present study the authors tested participants in a similar paradigm but asking participants to perform either individual eye or hand movements, or simultaneous hand+eye movements (in separate experiments). The pattern in the data is rather complex, with the modulation of saccadic peak velocity and modulation of reach movements seemingly absent in the dual-movement task (exp 1) and only a modulation of peak reach velocity being absent in single-movement task (exp 2).

Beside being roughly consistent with previous research showing a link between perceptual decision and eye movements, the take home message of the present study is, in my opinion, unclear. The discussion states that these findings show how interactions between eye and hand motor planning can occur also when these are made during unrelated perceptual decision (why would they not?), but do not provide mechanistic explanation for the results. Overall, I find that the importance and significance of this study is not communicated clearly. Additionally, the conclusions of the authors depend on interpreting absence of evidence (null NHST tests) as evidence for absence, however it's not obvious that these non-significant tests provide evidence for the null hypothesis (they may well not, given also that the sample size is relatively small, $n=11$ and $n=8$). Finally, there is also a potential confound (related to possible differences in response times between single and dual task

conditions) that is not addressed.

Below are some more specific comments:

- Was the mean response times (for either saccades and hand reaching) different in the single and dual experiment different? Currently only z-scored times are reported, so this is unclear. I would expect that these may be larger in the dual-task experiment (exp. 1) compared to the exp. 2, and their variance may be larger too. If that is the case, perhaps one complication in interpreting the current results is that the small interference effect due to the perceptual decision simply becomes harder to detect in such conditions?

- line 55-onward: the hypothesis seems vague, I think the authors need to do a better job in explaining the rationale and indicating what the hypothesis entails. What does it mean that perceptual decision would "prioritize" decision-irrelevant eye movements rather than reach movements? As I understand it, the effects of perceptual decisions on decision-irrelevant movements are considered a side-effects of multiplexing of signals in the neural circuits – a sort of interference. So what does it mean that the interference on a certain type of movements would be "prioritized" compared to another, and why would it be the case?

- related to the previous point, I am not aware of studies showing an effect of perceptual decision on decision irrelevant eye movements (e.g. similar to reference n. 9, the study by Joo, Katz & Huk, PNAS 2016) but for hand reaching movements; in fact unless I am missing something none of the references cited shows such an effect. In other words, there is no clear evidence that a similar interference effect occurs for hand movements, so why is it interesting/important to investigate the prioritization of interference effects on eye and hand movements? Note: I don't want to necessarily say that this is not interesting or important – it may well be – but I think the paper fails to convey why studying this is important/interesting. I think the authors need to do explain more clearly the rationale and scope of the study.

- Supplementary figure 1 is not very informative; it's very hard to tell what is going on with the data. To allow visual comparison of the data I recommend plotting each observer in a separate panel, with their individual functions plotted along the datapoints, using different colors for the same vs different conditions.

- line 120 "almost identical": it's hard to judge whether the functions are almost identical or not From Fig. 2 – it seems there could even be a slightly lower performance in the "Different" condition. Please provide statistical tests and use Bayes factors/equivalence tests to quantify more precisely support for the null hypothesis of no difference between conditions.

- Please provide Bayes factors or equivalence tests for null-results that are important for your conclusions. For example, it is said that coherence do not affect reach reaction times in the same, dual-movement condition, but judging from fig 2e there seems to be a trend, and indeed the p-value here is 0.07 so I doubt this can provide strong evidence in favour of the null hypothesis.

Minor:

- introduction: it should be acknowledged that whereas many studies showed how decisions influences ongoing hand movements, there is also evidence showing that faster movements like eye saccades, when used to communicate a perceptual decision, seems to be mostly planned *after* the decision is completed – see Lisi, Morgan & Solomon, 2022 (<https://doi.org/10.1038/s42003-022-03141-1>)

- There are some grammatical anomalies, please revise the manuscript to check english language. For instance, at line 45: should be "multiple decision-irrelevant" – in english determiners like multiple, many etc. come before adjectives and other noun modifiers

Reviewer #3 (Remarks to the Author):

Review of Matsumiya and Furukawa, Nat Comm Biol, 2023

The authors investigated how simultaneous saccades and reach movements are influenced by the formation of a perceptual decision in a motion discrimination task. They report that the latency of saccadic eye movements towards a decision-irrelevant target was systematically affected by the coherence (motion strength) of the decision target, yet reach movement latencies were not. These results were only observed when observers engaged in an active decision task, not when they passively viewed the same target, ruling out that the results could be an effect of any sort of motion-induced drift when viewing the target. Even though the overall logic of the experimental results is strong, I have concerns about the interpretation of the data due to high variability, small sample size, and over-reliance on statistical significance testing in multiple tests. On the upside, the paper is very well written and a pleasure to read.

Overall comments

- Approach and logic: at times, it is unclear what the relevance of understanding impact of perceptual decisions on irrelevant / unrelated motor action is. What is this a model of? How are decision-irrelevant action components relevant for our understanding of covert decision processes? It would be helpful to make this more explicit throughout the intro and discussion, especially considering that the main finding is very specific: an irrelevant action is only impacted if it is performed simultaneously with another decision-irrelevant action. Adding to the confusion is the fact that it seems that the overall question really is less about the impact of decision processes on irrelevant actions, but on which action is prioritized – the eye or the hand movement (l.74-75). It would be helpful if the authors clearly stated the relevance of each alternative outcome before summarizing their results.

- Interpretation of results: my main concern is with the variability in individual data and the fact that some subjects appear to be at chance performance even at the highest coherence level. How can such results be interpreted? See also more detailed comments re. statistical analysis and reporting below.

Specific (major and minor) comments

l.32: please consider replacing “generally” with “often”. Many recent studies (as in: published in the last 15 years) do not follow this simple serial model.

p.40: this is a nice summary and is consistent with literature on eye movements during decision making, see this recent review, which the authors could consider citing:

<https://pubmed.ncbi.nlm.nih.gov/35676097/>

Suppl. Figure 1 (and others): I’m intrigued by the fact that several subjects appear to perform at around chance even at the highest level of coherence. The variability in the data is enormous, likely owing to individual variabilities in motion sensitivity. Why did the authors choose a constant stimulus design (and why these particular coherence levels) rather than a threshold procedure, which would have accounted for this? As such, it seems that data for observers who perform at chance can hardly be interpreted.

Figure 2: fitted lines / model fits seem poor and are particularly misleading for reach reaction times. Why is a linear model used here?

l.143-144 and throughout: please report exact p-values. What were the mean saccade latencies here? All figures show standardized data, so it would be helpful to see actual means here.

l.151 following (also l.233 etc.): please clarify if these results were all obtained in separate F-tests / repeated-measures ANOVAS, and if yes, why were they not combined (i.e., factors “viewing condition” and “motion strength”)? Conducting multiple tests increases the risk of false positives. Also, the

authors solely focus on main effects and do not statistically evaluate interactions, yet, the data are interpreted as if there were interactions (effect on latencies in the active vs. passive task in saccades but not reaches). Moreover, some of the p-values are pretty borderline, yet are interpreted alongside much clearer results as indicating "no effect". Please consider supplying effect sizes so the reader can gauge the magnitude and meaning of these reported null effects.

l.286 following: it is not clear to me what part of the results this interpretation is based on: "We found that simultaneous decision-irrelevant saccade and reach movements suppress the modulation of saccade peak velocities, but not saccade reaction times, by perceptual decisions". Where is the evidence for an inhibitory process here? The authors observe differential effects on saccades and reaches, and differential effects on latency and peak velocity (though this has to be confirmed in a model that includes all these factors in one analysis). But this is not evidence of an inhibitory process per se.

Methods: even though this might not have been a concern with the overall finding, it would be helpful to report how stable fixation was during the presentation of the motion stimulus. This type of stimulus typically elicits strong drift or even pursuit. Was this the case here?

Responses to the reviewers' comments

General revisions:

In response to reviewer concerns about whether the actual reaction times differed between saccades and reaches and between the dual- and single-movement tasks and about whether proper statistical justification is provided for the present results, the revised manuscript now includes (1) analyses of the actual reaction times; (2) statistical evidence for the null hypothesis (Bayes factors and effect sizes); and (3) clarification of the logic of the Introduction and Discussion sections.

All revisions to the manuscript are highlighted in yellow.

Responses to Reviewer 1

1. **Reviewer's comment:** The results are generally clear. One missing analysis, in my opinion, is the actual reaction times in the different paradigms. For example, the lack of alteration in reach reaction time relative to saccade reaction time could reflect the fact that reach reactions times are much longer than saccade reaction times.

Response: Thank you for your comments. In accordance with your suggestion, we have changed the reaction times from the z-scores to the actual reaction times in the revised manuscript (see Fig. 2 for experiment 1 and Fig. 5 for experiment 2). We analysed the actual reaction times to examine whether reach reaction times are much longer than saccade reaction times in the present study. Reaction times for all conditions (e.g., viewing condition in the motion direction discrimination task, motion coherence and movement direction) were averaged for saccade and for reach movements. This result showed that reach reaction times were not significantly longer than saccade reaction times in experiment 1 [Fig. 6a; $F(1, 7) = 3.47$, $P = 0.10$], although the difference in reaction times between saccade and reach was 23.95 ms in experiment 1, as shown in Fig. 6a. This indicates that reach reaction times were not much longer than saccade reaction times. In addition, the standard deviation of reaction times was not significantly different between

Figure 2. Psychometric functions and reaction times in experiment 1. **(a)** Psychometric functions for the same and different tasks as a function of motion coherence. **(b and c)** Saccade reaction times for the same **(b)** and different **(c)** tasks as a function of motion coherence. **(d and e)** Reach reaction times for the same **(d)** and different **(e)** tasks as a function of motion coherence. Solid and dashed lines are the fitted lines for the active decision-making and passive viewing conditions, respectively. Results are the mean \pm standard error. $n = 8$.

Figure 5. Reaction times and peak velocities in experiment 2. (a–d) Reaction times for the saccade (a and c) and reach (b and d) movement conditions as a function of motion coherence. (a and b) The long (400 ms) duration condition. (c and d) The short (100 ms) duration condition. (e and f) Peak velocities for the saccade (e) and reach (f) movement conditions as a function of motion coherence. Results are the mean \pm standard error. $n = 11$.

saccade and reach [Fig. 6b; $F(1, 7) = 0.02$, $P = 0.89$], suggesting that the lack of alteration in reach reaction times with motion coherence cannot be explained by differences in reaction time variability. On the other hand, reach reaction times were significantly longer than saccade reaction times in experiment 2 [Fig. 6c; $F(1, 10) = 45.97$, $P < 0.01$], although the difference in reaction times between saccade and reach was 64.10 ms. The standard deviation of reaction times in experiment 2 was also significantly different between saccade and reach [Fig. 6d; $F(1, 10) = 31.26$, $P < 0.01$]. Nevertheless, reach reaction times were modulated by motion coherence in experiment 2. These results therefore rule out the possibility that the lack of the alteration in reach reaction time relative to saccade reaction time could reflect the fact that reach reaction times are much longer than saccade reaction times. We have added this information and these new data to the revised manuscript.

Figure 6. Comparison of reaction times and standard deviations between saccade and reach movements for each of experiments 1 and 2. **(a and c)** Reaction times. **(b and d)** Standard deviations. Results are the mean \pm standard error.

Main text, page 13, lines 250-265:

It may be a concern that the lack of alteration in the reach reaction time relative to the saccade reaction time reflected the fact that reach reaction times were much longer than saccade reaction times. We analysed the actual reaction times to examine whether reach reaction times are much longer than saccade reaction times. Reaction times for all conditions (i.e., viewing condition in the motion direction discrimination task, motion coherence and movement direction) were averaged for both saccade and reach movements. Reach reaction times were not significantly longer than saccade reaction times [Fig. 6a; $F(1, 7) = 3.47$, $P = 0.10$, $BF = 1.26$, $\eta_p^2 = 0.33$], although the difference in reaction times between saccade and reach was 23.95 ms in experiment 1, as shown in Fig. 6a. This indicates that reach reaction times were not much longer than saccade reaction times. In addition, the standard deviation of reaction times was not significantly different between saccade and reach [Fig. 6b; $F(1, 7) = 0.02$, $P = 0.89$, $BF = 0.43$, $\eta_p^2 = 0.0028$], suggesting that the lack of alteration in reach reaction times with motion coherence cannot be explained by differences in reaction time variability. Thus, these results rule out the possibility that the lack of alteration in reach reaction time relative to saccade reaction time could reflect the fact that reach reaction times are much longer than saccade reaction times.

Main text, page 21, line 306 to page 16, line 314:

To examine the difference in reaction times between saccade and reach movements, we compared saccade reaction times in the saccade-only task and reach reaction times in the reach-only task. Reaction times for all conditions (i.e., motion coherence and duration) were averaged for both saccade and reach movements. Reach reaction times were significantly longer than saccade reaction times [Fig. 6c; $F(1, 10) = 45.97$, $P = 4.86 \times 10^{-5}$, $\eta_p^2 = 0.82$]. The standard deviation of reaction times was also significantly different between saccade and reach [Fig. 6d;

$F(1, 10) = 31.26, P = 0.00023, \eta_p^2 = 0.76$]. Nevertheless, not only saccade reaction times, but also reach reaction times were modulated by motion coherence in experiment 2.

2. **Reviewer's comment:** line 43 (and abstract): I think it would be helpful in both the abstract and also early introduction to define and clarify things a bit more. In particular, I would recommend clarifying what is exactly meant by terms like “decision-irrelevant” and “interact”. For example, in lines 17-18 of the abstract, I'm not sure what the difference is between “motor actions must be related to decision formation” and “perceptual decisions also influence a motor action unrelated to decisions”? Isn't the first already implying the second? I know that you probably want to say something else, but it needs to be crystal clear very early on in the paper. The introduction does a slightly better job of first saying that sometimes, one might think that motor actions are just an output stage, but in reality there might be reach curvatures etc that reflect the internal processing. But, again, the formulation of the entire thing needs to be much more clear. Returning back to line 43, I again see a statement like “eye movements that are decision irrelevant”, but I still do not understand what is meant. I am sure it will get clearer later in the paper, but why leave it a mystery? Clarify as early as possible.

Response: Thank you for your comment. We apologise for the lack of clarity in the difference between ‘motor actions must be related to decision formation’ and ‘perceptual decisions also influence a motor action unrelated to decisions’ in the Abstract of the previous version of the manuscript. We have revised the text to state that actions are influenced by perceptual decisions even when the action is irrelevant to the perceptual decision-making task.

In addition, you asked us to clarify, as early as possible, what is meant by the term ‘interact’. In response to your suggestion, we have added the following text early in the Introduction of the revised manuscript: ‘motor actions are continuously affected by ongoing judgements that are not yet complete, suggesting an interaction between perceptual judgements and motor actions’.

Furthermore, to improve the formulation of the manuscript, particularly the Introduction, we have changed the structure of the paragraphs early in the Introduction of the revised manuscript. In the second paragraph of the Introduction, we have described an interaction between perceptual decisions and motor actions using reach movement tasks as examples. Then, in the third paragraph, we have revised the text to state that ‘the trajectories of saccadic eye movements also elicit systematic deviations in saccade curvature and endpoints when saccadic eye movements are used to report judgements in a perceptual judgement task’. We believe that these revisions clarify what is meant by the terms such as ‘decision-irrelevant’ and ‘interact’.

Abstract, page 2, lines 16-19:

Although intuition suggests that these actions must be related to the formation of decisions about where to move body parts, actions have been reported to be influenced by perceptual judgements even when the action is irrelevant to the perceptual judgement.

Main text, page 3, lines 34-35:

For example, saccadic eye movements are made after a decision about where to move the eyes based on sensory information.

Main text, page 3, lines 37-38:

suggesting an interaction between perceptual judgements and motor actions

Main text, page 3, lines 43-44:

suggesting a continuous interaction between perceptual judgements and reach movements

Main text, page 3, lines 45 to page 4, line 52:

the trajectories of saccadic eye movements also elicit systematic deviations in saccade curvature and endpoints when saccadic eye movements are used to report judgements in a perceptual judgement task⁹, indicating that oculomotor output can also be continuously affected by ongoing perceptual judgements. A recent study has demonstrated that saccades are not yet ready to launch when perceptual decision processes terminate¹⁰, suggesting that perceptual decisions and oculomotor responses rely on temporally distinct streams of evidence. These findings imply a continuous interaction between perceptual judgements and saccadic eye movements¹¹, like reach movements.

3. **Reviewer's comment:** line 45: I guess one problem I have is that the actions themselves involve decisions. So, when you say decision-irrelevant actions, I get confused. I think it is best to avoid this word decision as much as possible, and just say that subjects have to do a dual task: one perceptual judgement (or whatever else it is) and the other is a reach or eye movement. And, we show that the perceptual judgement interferes with the motor action even if the motor action is irrelevant to the judgement.

Response: Thank you for making us aware that the actions themselves involve decisions and that this causes confusion. In accordance with your suggestion, we have changed the word 'decision' to 'judgement' as necessary throughout the revised manuscript.

Abstract, page 2, lines 14-26:

Perceptual judgements are formed....

Main text, page 3, line 29 to page 7, line 136:

Studies of perceptual judgements depend on the ability.... Perceptual judgements

and.... In a perceptual judgement task,

4. **Reviewer's comment:** P. 7, line 56: what does it mean for the perceptual decision to prioritize decision-irrelevant eye movements? What does "prioritise" mean???? I think the message is that eye movements experience larger interference from perceptual decisions than reach movements. This is much more descriptive and appropriate in my opinion.

Response: Thank you for your comment. Your interpretation of the meaning of 'prioritise' is correct. In accordance with your suggestion, we have revised the text to state that perceptual judgements interfere more with eye movements than with reach movements.

Main text, page 5, lines 84-85:

perceptual judgements may interfere more with eye movements than with reach movements

5. **Reviewer's comment:** line 61: add one sentence clarifying after the first sentence what you looked for. e.g. "We then looked for perturbations in reach or eye movement trajectory..." (or whatever else you measured...at this stage of the introduction, I still do not know what you measured yet).

Response: In response to your suggestion, we have added a sentence clarifying what we looked for at that location. Thank you for the advice.

Main text, page 6, lines 112-113:

We then looked for perturbations in the reaction times and peak velocities of eye or reach movements.

6. **Reviewer's comment:** lines 62-64: now, suddenly the text says “eye and reach movements *affect* concurrent perceptual processes”. Isn't it the perceptual process that interferes with the movement, at least up to this point in the abstract and introduction? I think the problem is that there is too much left to the reader in the text. It wouldn't hurt to explain paradigms and experiments rather than just list citations. This way, confusions like mine at this sentence would be minimised.

Response: We apologize for the confusion and we appreciate your comment. In accordance with your suggestion, we have added a description of the previous paradigms, in addition to the citations.

Main text, page 6, lines 114 to page 7, line 117:

For example, in the previous paradigms, participants discriminated changes in visual targets during eye movements²¹⁻²⁴ or reach movements²⁵. In other examples, participants discriminated briefly presented visual patterns during eye and reach movements^{26, 27} or judged the location of body parts during eye and reach movements^{28, 29}.

7. **Reviewer's comment:** lines 66-75 are very nice

Response: Thank you.

8. **Reviewer's comment:** line 120: can you assess this similarity of psychometric curves statistically? I think that this is critical to show.

Response: In response to your suggestion, we have statistically assessed the similarities of the psychometric curves.

Main text, page 9, line 172:

The psychometric functions were almost identical between the same and different

tasks [$F(1, 14) = 0.58, P = 0.46, BF$ (Bayes factor) = 1.04, $\eta_p^2 = 0.54$],

9. **Reviewer's comment:** line 134: you might want to also look at Buonocore et al, 2017, 2021 providing a mechanistic explanation for why kinematics might be altered with sensory interference in the citation that you made

Response: Thank you for the information on these interesting and relevant articles. We have cited these papers in the revised manuscript, accordingly.

Main text, page 10, lines 186-187:

there is also evidence that saccade peak velocities reflect the degree of certainty with which a perceptual decision is made (i.e., confidence in a decision)^{32, 35, 36}.

10. **Reviewer's comment:** line 157: One big difference between reaches and saccades is that the reaches probably have 3x the reaction times of the saccades. So, the lack of effect could simply be because there was too much time between the motion patch and the reach. You do not show the actual manual and saccadic reaction times, but I think you should show them. And, you should consider the possibility that the large difference in reaction time between saccades and reaches could explain your results.

Response: As we stated in our response to your first comment above (Reviewer comment #1), we have shown the actual saccade and reach reaction times in the revised manuscript (see Figs. 2 and 5). In addition, we have analysed the actual reaction times to examine whether reach reaction times are much longer than saccade reaction times (see Fig. 6). As shown in Fig. 6, the difference in reaction times between saccade and reach was 23.95 ms in experiment 1 and 64.10 ms in experiment 2. These results indicated that the reaches did not have triple the reaction times of the saccades. In addition, reaction times for all conditions (viewing condition in the motion direction discrimination task, motion coherence and movement direction) were averaged for saccade and for reach movements, and

the difference in their average reaction times between saccades and reaches was analysed for both experiments 1 and 2. For experiment 1, reach reaction times were not significantly longer than saccade reaction times [Fig. 6a; $F(1, 7) = 3.47$, $P = 0.10$]. The standard deviation of reaction times was not significantly different between saccade and reach [Fig. 6b; $F(1, 7) = 0.02$, $P = 0.89$]. For experiment 2, reach reaction times were significantly longer than saccade reaction times [Fig. 6c; $F(1, 10) = 45.97$, $P < 0.01$]. The standard deviation of reaction times was also significantly different between saccade and reach [Fig. 6d; $F(1, 10) = 31.26$, $P < 0.01$]. Nevertheless, not only saccade reaction times, but also reach reaction times were modulated by motion coherence in experiment 2. Thus, these results rule out the possibility that the large difference in reaction time between saccade and reach may explain the present results. We have added these descriptions and these new data to the revised manuscript.

See the response to the first comment above for corrected text in the revised manuscript:

Main text, page 13, lines 250-265;

Main text, page 15, line 306 to page 16, line 314

11. **Reviewer's comment:** line 160: I think I expected this result (see comment above about the kinematic alterations)

Response: Thank you.

12. **Reviewer's comment:** line 195: again, I cannot see whether they are “nearly identical” because they are put in different panels. Some statistical quantification and/or plotting in the same panel is necessary

Response: In response to your suggestion, we have statistically assessed the similarities of the psychometric curves and have plotted the psychometric functions

in the same panel for each duration (see Fig. 4).

Main text, page 14, lines 282-284:

The psychometric functions were nearly identical in the saccade-only and reach-only tasks [$F(1, 10) = 1.77, P = 0.21, BF = 0.34, \eta_p^2 = 0.15$ for 100-ms duration; $F(1, 10) = 0.00, P = 0.99, BF = 0.20, \eta_p^2 = 0.00$ for 400-ms duration],

Main text, page 42, Figure 4:

Figure 4. Psychometric functions for the short (100 ms; **a**) and long (400 ms; **b**) duration conditions as a function of motion coherence in experiment 2. (**a**) Bright blue and yellow lines are the fitted lines for the saccade and reach movement conditions, respectively. (**b**) Dark blue and brown lines are the fitted lines for the saccade and reach movement conditions, respectively. Results are the mean \pm standard error. $n = 11$.

13. **Reviewer's comment:** line 209: but maybe reach reaction time in the dual task was longer than in the single task. i.e. when required to both reach and look, the reaction times are longer than with only reaching. Similarly for saccades.

Response: We have analysed the difference in reaction time between the dual and single tasks (Fig. 6a and c), and a two-way analysis of variance was performed with effectors and tasks as factors. There was a significant interaction between

effector and task [$F(1, 17) = 6.65, P < 0.05$]. Reach reaction times were significantly shorter in the dual task than in the single task [$F(1, 17) = 76.11, P < 0.01$]. The same was true for saccades [$F(1, 17) = 66.93, P < 0.01$]. Thus, these results rule out the possibility that the lack of the alteration in the reach reaction time in the dual task could reflect the fact that reach reactions times are much longer in the dual task than in the single task.

Main text, page 16, lines 315-325:

A possible concern is that the lack of alteration in the reach reaction time in the dual task (experiment 1) reflected the fact that reach reaction times were much longer in the dual task (experiment 1) than in the single task (experiment 2). We analysed the difference in reaction times between the dual and single tasks (Fig. 6a and c). A two-way analysis of variance (ANOVA) was performed with effectors and tasks as factors. There was a significant interaction between effector and task [$F(1, 17) = 6.65, P = 0.020, \eta_p^2 = 0.28$]. Reach reaction times were significantly shorter in the dual task than in the single task [$F(1, 17) = 76.11, P = 1.10 \times 10^{-7}, \eta_p^2 = 0.82$]. The same was true for saccades [$F(1, 17) = 66.93, P = 2.69 \times 10^{-7}, \eta_p^2 = 0.80$]. Thus, these results rule out the possibility that the lack of alteration in the reach reaction time in the dual task could reflect the fact that reach reactions times are much longer in the dual task than in the single task.

Responses to Reviewer 2

1. **Reviewer's comment:** Beside being roughly consistent with previous research showing a link between perceptual decision and eye movements, the take home message of the present study is, in my opinion, unclear. The discussion states that these findings shows how interactions between eye and hand motor planning can occur also when these are made during unrelated perceptual decision (why would they not?), but do not provide mechanistic explanation for the results. Overall, I find that the importance and significance of this study is not communicated clearly. Additionally, the conclusions of the authors depends on interpreting absence of evidence (null NHST tests) as evidence for absence, however it's not obvious that these non-significant tests provide evidence for the null hypothesis (they may well not, given also that the sample size is relatively small, $n=11$ and $n=8$). Finally, there is also a potential confounds (related to possible differences in response times between single and dual task conditions) that is not addressed.

Response: Thank you for your comments. In accordance with your suggestion, we have provided a mechanistic explanation for the present results in the revised manuscript. To explain our results in a mechanistic way, we have used the channel modulation hypothesis (Pesaran et al., 2021). Based on this hypothesis, we created a channel modulation model (see Fig. 7). In this model, perceptual decision making consists of the three separate systems in parallel. Each of these systems forms the communication channel to the motor system. Activity in a system of perceptual decisions drives responses in the motor system. In addition, a modulator network exists between the perceptual decision-making system and the motor system. The modulator network is triggered by motor commands from eye and/or reach movements, and activity in the modular network can change the motor system response to input from the perceptual decision system, modulating the communication channel. For example, when simultaneous eye and reach movements are made, the modulator network closes the channel from the system guiding saccade peak velocities to the motor system and the channel from the system guiding reach reaction times to the motor system (Fig. 7b). As a result, the

Figure 7. Channel modulation model. **(a)** Perceptual decision making comprises three separate systems in parallel. Each of these systems forms the communication channel to the motor system. Activity in a system of perceptual decisions drives responses in the motor system. The modulator network is triggered by motor commands from saccade and/or reach movements, and activity in the modular network can change the motor system response to input from the perceptual decision system. **(b)** Channel modulation during simultaneous saccades and reaches. **(c)** Channel modulation during saccades. **(d)** Channel modulation during reaches. **(e)** The three separate systems of perceptual decision making are related to the expected reward or value associated with a perceptual decision and the ability to have confidence in a perceptual decision.

perceptual decision-making process only affects saccade reaction times (see the Discussion section in the revised manuscript for details). Thus, the present study suggests that the concept of communication channel modulation between the perceptual decision-making system and the motor system may be the key to understanding the mechanisms underlying the link between decision-related activity and decision-unrelated motor processes.

In addition, in response to your suggestion, we have used Bayes factors/equivalence tests to quantify support for the null hypothesis of no significant difference between conditions. For more information, please see our response to Reviewer comment #7 below.

Finally, we have analysed the differences in response times between the single- and dual-task conditions. For more information, please see our response to Reviewer comment #2 below.

Main text, page 24, line 490 to page 25, line 524:

To provide a mechanistic explanation for these complicated results, we consider the channel modulation hypothesis¹⁹. According to this hypothesis, the communication channel is formed from projections from the perceptual decision-making system to the motor system (Fig. 7a). As a result, activity in the perceptual decision-making system affects responses in the motor system. In addition, a modulator network is placed between the perceptual decision-making system and the motor system. The modulator network works through motor commands from eye and/or reach movements. Activity in the modulator network can alter the motor system response to input from the perceptual decision-making system, modulating the communication channel. Such a channel modulation can produce changes in reaction times and peak velocities for movements by opening the channel that communicates perceptual decision signals to guide a motor response (e.g., saccades) or by closing the channel that communicates perceptual decision signals to guide a different motor response (e.g., reaches).

The present complicated results can be explained by making two assumptions regarding the channel modulation hypothesis (Fig. 7a). First, we assume that perceptual decision-making occurs in three separate systems: (i) the system guiding saccade reaction times, (ii) the system guiding reach reaction times, and (iii) the system guiding saccade peak velocities. Therefore, three communication channels are formed from projections from each of these three systems to the motor system. Second, we assume that a modulator network operates differently depending on the movement task. When simultaneous eye and reach movements are made, the modulator network closes two of the three communication channels: the channel from the system guiding saccade peak velocities to the motor system, and the channel from the system guiding reach reaction times to the motor system (Fig. 7b). As a result, the perceptual decision-making process only affects saccade reaction times. When a saccade without a reach is made, the modulator network closes the channel from the system guiding reach reaction times to the motor system (Fig. 7c). As a result, the perceptual decision-making process affects the reaction times and peak velocities of the saccade. When a reach without a saccade is made, the modulator network closes two channels: the channel from the system guiding saccade reaction times to the motor system, and the channel from the system guiding saccade peak velocities to the motor system (Fig. 7d). As a result, the perceptual decision-making process affects reach reaction times.

In addition, we speculate that the three separate systems of perceptual decision making described above may be related to the expected reward or value associated with a judgement and the ability to have confidence in a judgement (Fig. 7a and e). Saccade and reach reaction times have been reported to reflect the expectation of the reward or value associated with the movement target^{33, 34}. Meanwhile, saccade peak velocities have been reported to reflect the degree of certainty with which a perceptual decision is made (i.e., confidence in a decision)³², although an influence on saccade peak velocities from the reward or value associated with the saccade target has been demonstrated in many studies⁴⁰⁻⁴². Given that reaction time and peak velocity reflect different features of perceptual decisions, our findings suggest

that simultaneous saccade and reach movements may be involved in suppressing the channel related to decision confidence in the oculomotor circuitry and the channel related to the expectation of a reward or the value in the manual circuitry, and may interact only with the channel related to the expectation of a reward or the value in the oculomotor circuitry. Thus, the concept of communication channel modulation between the perceptual decision-making system and the motor system may be a key to understanding the mechanisms underlying the link between decision-related activity and decision-unrelated motor processes.

2. **Reviewer's comment:** Was the mean response times (for either saccades and hand reaching) different in the single and dual experiment different? Currently only z-scored times are reported, so this is unclear. I would expect that these may be larger in the dual-task experiment (exp. 1) compared to the exp. 2, and their variance may be larger too. If that is the case, perhaps one complication in interpreting the current results is that the small interference effect due to the perceptual decision simply becomes harder to detect in such conditions?

Response: Thank you for your comment. Yes, the mean response times differed between the single- and dual-task experiments for both saccades and reaches (Fig. 6). Surprisingly, however, contrary to your expectation, the mean response times were significantly larger in the single-task experiment than in the dual-task experiment for both saccades and reaches [$F(1, 17) = 66.93, P < 0.01$ for saccades; $F(1, 17) = 76.11, P < 0.01$ for reaches]. On the other hand, the mean standard deviations of reaction times were not significantly different between the dual- and single-task experiments for both saccades and reaches [$F(1, 17) = 1.72, P = 0.21, BF < 0.73$ for saccades; $F(1, 17) = 2.61, P = 0.12, BF < 0.97$ for reaches]. Thus, these results rule out the possibility that the small interference effect due to the perceptual decision simply becomes harder to detect in the dual-task experiment compared with the single-task experiment. We have added these descriptions to the revised manuscript.

Figure 6. Comparison of reaction times and standard deviations between saccade and reach movements for each of experiments 1 and 2. (a and c) Reaction times. (b and d) Standard deviations. Results are the mean \pm standard error.

Main text, page 16, line 315 to page 17, line 338:

A possible concern is that the lack of alteration in the reach reaction time in the dual task (experiment 1) reflected the fact that reach reaction times were much longer in the dual task (experiment 1) than in the single task (experiment 2). We analysed the difference in reaction times between the dual and single tasks (Fig. 6a and c). A two-way analysis of variance (ANOVA) was performed with effector and tasks as factors. There was a significant interaction between effector and task [$F(1, 17) = 6.65, P = 0.020, \eta_p^2 = 0.28$]. Reach reaction times were significantly shorter in the dual task than in the single task [$F(1, 17) = 76.11, P = 1.10 \times 10^{-7}, \eta_p^2 = 0.82$]. The same was true for saccades [$F(1, 17) = 66.93, P = 2.69 \times 10^{-7}, \eta_p^2 =$

0.80]. Thus, these results rule out the possibility that the lack of alteration in the reach reaction time in the dual task could reflect the fact that reach reaction times are much longer in the dual task than in the single task.

Another concern may be that the lack of alteration in the reach reaction time in the dual task (experiment 1) reflected the fact that the variance of reach reaction times was larger in the dual task (experiment 1) than in the single task (experiment 2). We analysed the difference in the mean standard deviation of reaction times between the dual and single tasks (Fig. 6b and d). A two-way ANOVA was performed with effectors and tasks as factors. There was a significant interaction between effector and task [$F(1, 17) = 14.60, P = 0.0014, \eta_p^2 = 0.46$]. The mean standard deviation of reach reaction times was not significantly different between the dual and single tasks [$F(1, 17) = 2.61, P = 0.12, BF < 0.97, \eta_p^2 = 0.13$]. Moreover, the mean standard deviation of saccade reaction times was not significantly different between the dual and single tasks [$F(1, 17) = 1.72, P = 0.21, BF < 0.73, \eta_p^2 = 0.092$]. Thus, these results rule out the possibility that the lack of alteration in the reach reaction time in the dual task reflected the fact that the variance of reach reaction times was larger in the dual task than in the single task.

3. **Reviewer's comment:** line 55-onward: the hypothesis seems vague, I think the authors need to do a better job in explaining the rationale and indicating what the hypothesis entails. What does it mean that perceptual decision would “prioritize” decision-irrelevant eye movements rather than reach movements? As I understand it, the effects of perceptual decisions on decision-irrelevant movements are considered a side-effect of multiplexing of signals in the neural circuits – a sort of interference. So what does it mean that the interference on a certain type of movements would be “prioritized” compared to another, and why would it be the case?

Response: Thank you for your comment. We now recognise that what we meant by ‘prioritise’ was unclear. Another reviewer also highlighted the same issue. As you pointed out, what we would like to say by ‘prioritise’ is that perceptual

judgements more strongly interfere with eye movements than with reach movements when simultaneous eye and reach movements are made. To avoid confusion due to the use of the term ‘prioritise’, we have used the term ‘interfere’ instead of ‘prioritise’ in the revised manuscript. In addition, we have modified the manuscript to more clearly state the hypothesis. We agree with your argument that the effects of perceptual decisions on decision-irrelevant movements are considered side effects of the multiplexing of signals in the neural circuits. Related to our paper, previous research has provided evidence that decision-making-related activity arises in both oculomotor and manual brain areas such as the lateral intraparietal (LIP) area and the medial intraparietal (MIP) area, respectively (de Lafuente et al., 2015). Given that perceptual judgement-related activity in the oculomotor system interferes with judgement-irrelevant eye movements (Joo et al., 2016), it seems reasonable that we hypothesise that perceptual judgement-related activity in the manual system may also interfere with judgement-irrelevant reach movements when reach movements are made without eye movements. In addition, previous research has shown that the activity of MIP neurons is greatly attenuated when perceptual judgements are communicated by eye movements while LIP neurons still activate when perceptual judgements are communicated by reach movements, as well as eye movements (de Lafuente et al., 2015). We believe that these findings provide the rationale for our hypothesis that perceptual judgements may interfere only with eye movements when simultaneous judgement-irrelevant eye and reach movements are made. We have added these descriptions to the revised manuscript.

Main text, page 4, line 70 to page 6, line 94:

However, it is not known how the signal interference occurs in dual-task paradigms such as simultaneous eye and reach movements. Such paradigms offer the opportunity to investigate how perceptual judgement-related signals flow between motor systems, which helps to explain the mechanisms of communication across

motor systems¹⁹ during interference between perceptual decision making and motor actions. A previous neurophysiological study showed that perceptual judgement-related activity arises in both oculomotor and manual brain areas such as the LIP and MIP, respectively¹⁷. Importantly, that neurophysiological study also showed that the activity of MIP neurons is greatly attenuated when perceptual judgements are communicated by eye movements (i.e., due to reduced perceptual judgement-related signals from MIP neurons, these signals likely interfere less with reach movements when eye movements are made) while LIP neurons still activate when perceptual judgements are communicated by reach movements, as well as eye movements (i.e., perceptual judgement-related signals from LIP neurons can still interfere with eye movements when reach movements are made)¹⁷. Therefore, we hypothesised that perceptual judgements may interfere more with eye movements than with reach movements when simultaneous judgement-irrelevant eye and reach movements are made (hypothesis 1).

Furthermore, it is not known whether perceptual judgements interfere with judgement-irrelevant reach movements without eye movements. Perceptual judgement-related activity arises in the LIP^{1, 12, 17, 20}, and such activity interferes with judgement-irrelevant eye movements¹⁸. Given that the MIP shows selectivity for both perceptual judgement-related activity and motor processes, similar to the LIP¹⁷, we hypothesised that perceptual judgement-related activity might interfere with judgement-irrelevant reach movements when reach movements are made without eye movements (hypothesis 2).

4. **Reviewer's comment:** related to the previous point, I am not aware of studies showing an effect of perceptual decision on decision irrelevant eye movements (e.g. similar to reference n. 9, the study by Joo, Katz & Huk, PNAS 2016) but for hand reaching movements; in fact unless I am missing something none of the references cited shows such an effect. In other words, there is no clear evidence that a similar interference effect occurs for hand movements, so why is it interesting/important to investigate the

prioritization of interference effects on eye and hand movements? Note: I don't want to necessarily say that this is not interesting or important – it may well be – but I think the paper fails to convey why studying this is important/interesting. I think the authors need to do explain more clearly the rationale and scope of the study.

Response: Thank you for raising this important point. As you pointed out, there is no evidence that a similar interference effect occurs for reach movements. In our manuscript, therefore, we considered it necessary to state the hypothesis that a similar interference effect would occur for reach movements. To derive this hypothesis, we introduced a neurophysiological study (de Lafuente et al., 2015) showing that perceptual judgement-related activity arises not only in oculomotor brain areas (e.g., LIP), but also in manual brain areas (e.g., MIP). Given that perceptual judgement-related activity interferes with judgement-irrelevant eye movements (Joo et al., PNAS, 2016), we hypothesised that perceptual judgement-related activity might also interfere with judgement-irrelevant reach movements when reach movements are made without eye movements. Importantly, the study by de Lafuente et al. has also provided evidence that the activity of MIP neurons is greatly attenuated when perceptual judgements are communicated by eye movements while LIP neurons still activate when perceptual judgements are communicated by reach movements, as well as eye movements. This finding raises an important question of whether perceptual judgements interfere with both judgement-irrelevant eye and reach movements or whether perceptual judgements interfere only with eye movements, even when simultaneous judgement-irrelevant eye and reach movements are made. We have added this information to the revised manuscript.

Main text, page 5, line 87 to page 6, line 109:

Furthermore, it is not known whether perceptual judgements interfere with judgement-irrelevant reach movements without eye movements. Perceptual judgement-related activity arises in the LIP^{1, 12, 17, 20}, and such activity interferes

with judgement-irrelevant eye movements¹⁸. Given that the MIP shows selectivity for both perceptual judgement-related activity and motor processes, similar to the LIP¹⁷, we hypothesised that perceptual judgement-related activity might interfere with judgement-irrelevant reach movements when reach movements are made without eye movements (hypothesis 2).

If we can obtain results that support hypotheses 1 and 2, these results would demonstrate that perceptual judgement-related signals continuously flow into the oculomotor system alone when multiple judgement-irrelevant actions are performed. Testing of hypothesis 1 would reveal whether simultaneous eye and reach movements are necessary for perceptual judgements to interfere only with judgement-irrelevant eye movements. However, even if we obtain results that support hypothesis 1, perceptual judgements might not interfere with judgement-irrelevant reach movements regardless of simultaneous eye and reach movements. To address this issue, we need to test hypothesis 2. Testing of hypothesis 2 would reveal whether interference between perceptual judgements and judgement-irrelevant motor actions is observed in reach movements without eye movements. Therefore, the fact that both hypotheses 1 and 2 are true would suggest that perceptual judgements interfere more with eye movements than with reach movements when simultaneous judgement-irrelevant eye and reach movements are made. These results will provide clues for understanding the mechanisms of communication across motor systems during perceptual decision making¹⁹.

5. **Reviewer's comment:** Supplementary figure 1 is not very informative; it's very hard to tell what is going on with the data. To allow visual comparison of the data I recommend plotting each observer in a separate panel, with their individual functions plotted along the datapoints, using different colors for the same vs different conditions.

Response: I apologize that Supplementary Fig. 1 was not very informative. In accordance with your suggestion, we have plotted each participant in a separate panel with their individual functions plotted along the datapoints using different

colours for the same and different conditions.

Supplementary Information, page 2, Supplementary figure 1:

6. **Reviewer’s comment:** line 120 “almost identical”: it’s hard to judge whether the functions are almost identical or not From Fig. 2 – it seems there could even be a slightly lower performance in the “Different” condition. Please provide statistical tests and use Bayes factors/equivalence tests to quantify more precisely support for the null hypothesis of no difference between conditions.

Response: To make it easier to judge whether the functions are almost identical or not, we have plotted the data for the same and different conditions in the same panel (see Fig. 2a). In addition, in accordance with your suggestion, we have applied repeated-measures analysis of variance (ANOVA) and Bayes

factors/equivalence tests to these data. The ANOVA showed that there was no significant difference between the same and different conditions [$F(1, 14) = 0.58$, $P = 0.46$] while the Bayes factors/equivalence tests showed that the main effect of condition (same vs. different) was invalid ($BF = 1.04$). We have added the results of these statistical tests to the revised manuscript.

Main text, page 9, lines 171-172:

The psychometric functions were almost identical between the same and different tasks [$F(1, 14) = 0.58$, $P = 0.46$, BF (Bayes factor) = 1.04, $\eta_p^2 = 0.54$],

Main text, page 40, Figure 2a:

7. **Reviewer's comment:** Please provide Bayes factors or equivalence tests for null-results that are important for your conclusions. For example, it is said that coherence does not affect reach reaction times in the same, dual-movement condition, but judging from fig 2e there seems to be a trend, and indeed the p-value here is 0.07 so I doubt this can provide strong evidence in favour of the null hypothesis.

Response: We have provided Bayes factors/equivalence tests for null results in the revised manuscript.

Main text, page 9, lines 171-172:

The psychometric functions were almost identical between the same and different tasks [$F(1, 14) = 0.58, P = 0.46, BF$ (Bayes factor) = 1.04, $\eta_p^2 = 0.54$],

Main text, page 10, lines 199-202:

motion coherence no longer modulated saccade reaction times in the passive viewing condition (open symbols in Fig. 2b and c) [same task, $F(4, 28) = 1.47, P = 0.24, BF < 1.81, \eta_p^2 = 0.17$; different task, $F(4, 28) = 0.43, P = 0.79, BF < 0.62, \eta_p^2 = 0.44$].

Main text, page 11, lines 207-215:

There were no significant main effects of decision-making condition (active and passive) [same task, $F(1, 7) = 0.30, P = 0.59, BF = 0.52, \eta_p^2 = 0.041$; different task, $F(1, 7) = 0.68, P = 0.41, BF = 0.71, \eta_p^2 = 0.088$]. There were no significant main effects of motion coherence level [same task, $F(4, 28) = 0.68, P = 0.61, BF = 0.071, \eta_p^2 = 0.088$; different task, $F(4, 28) = 0.70, P = 0.60, BF = 0.088, \eta_p^2 = 0.090$]. There were also no significant interactions between the decision-making condition and motion coherence level [same task, $F(4, 28) = 0.44, P = 0.78, BF = 0.13, \eta_p^2 = 0.059$; different task, $F(4, 28) = 0.19, P = 0.94, BF = 0.12, \eta_p^2 = 0.027$].

Main text, page 11, line 220 to page 12, line 227:

There were no significant main effects of decision-making condition (active and passive) [same task, $F(1, 7) = 0.0, P = 0.99, BF = 0.23, \eta_p^2 = 0.0006$; different task, $F(1, 7) = 0.40, P = 0.55, BF = 0.42, \eta_p^2 = 0.054$]. There were no significant main effects of motion coherence level [same task, $F(4, 28) = 0.78, P = 0.55, BF = 0.093, \eta_p^2 = 0.10$; different task, $F(4, 28) = 1.21, P = 0.33, BF = 0.12, \eta_p^2 = 0.15$]. There were also no significant interactions between the decision-making condition and motion coherence level [same task, $F(4, 28) = 1.64, P = 0.19, BF =$

0.72, $\eta_p^2 = 0.19$; different task, $F(4, 28) = 0.23$, $P = 0.92$, $BF = 0.13$, $\eta_p^2 = 0.031$].

Main text, page 12, lines 233-244:

For the same task, there was no significant main effect of decision-making condition (active and passive) [$F(1, 7) = 3.39$, $P = 0.11$, $\eta_p^2 = 0.33$], but Bayes factor analysis showed that reach peak velocities were substantially smaller in the active condition than in the passive condition [$BF = 3714.28$]. For the different task, there was no significant main effect of decision-making condition (active and passive) [$F(1, 7) = 0.51$, $P = 0.50$, $\eta_p^2 = 0.067$], and Bayes factor analysis also showed that the main effect of decision-making condition was not valid [$BF = 0.66$]. There were no significant main effects of motion coherence level [same task, $F(4, 28) = 2.01$, $P = 0.12$, $BF = 0.097$, $\eta_p^2 = 0.22$; different task, $F(4, 28) = 1.23$, $P = 0.32$, $BF = 0.10$, $\eta_p^2 = 0.15$]. There were also no significant interactions between the decision-making condition and motion coherence level [same task, $F(4, 28) = 1.02$, $P = 0.41$, $BF = 0.14$, $\eta_p^2 = 0.13$; different task, $F(4, 28) = 0.92$, $P = 0.47$, $BF = 0.20$, $\eta_p^2 = 0.12$].

Main text, page 13, lines 256-261:

Reach reaction times were not significantly longer than saccade reaction times [Fig. 6a; $F(1, 7) = 3.47$, $P = 0.10$, $BF = 1.26$, $\eta_p^2 = 0.33$], although the difference in reaction times between saccade and reach was 23.95 ms in experiment 1, as shown in Fig. 6a. This indicates that reach reaction times were not much longer than saccade reaction times. In addition, the standard deviation of reaction times was not significantly different between saccade and reach [Fig. 6b; $F(1, 7) = 0.02$, $P = 0.89$, $BF = 0.43$, $\eta_p^2 = 0.0028$],

Main text, page 14, lines 282-284:

The psychometric functions were nearly identical in the saccade-only and reach-

only tasks [$F(1, 10) = 1.77, P = 0.21, BF = 0.34, \eta_p^2 = 0.15$ for 100-ms duration; $F(1, 10) = 0.00, P = 0.99, BF = 0.20, \eta_p^2 = 0.00$ for 400-ms duration],

Main text, page 15, lines 295-297:

However, there were no significant interactions between duration and motion coherence [saccade, $F(4, 40) = 1.52, P = 0.21, BF = 0.11, \eta_p^2 = 0.13$; reach, $F(4, 40) = 0.81, P = 0.53, BF = 0.095, \eta_p^2 = 0.075$].

Main text, page 16, line 332 to page 17, line 336:

The mean standard deviation of reach reaction times was not significantly different between the dual and single tasks [$F(1, 17) = 2.61, P = 0.12, BF < 0.97, \eta_p^2 = 0.13$]. Moreover, the mean standard deviation of saccade reaction times was not significantly different between the dual and single tasks [$F(1, 17) = 1.72, P = 0.21, BF < 0.73, \eta_p^2 = 0.092$].

Main text, page 17, lines 346-349:

There was no significant difference in saccade peak velocity between the two durations (100 and 400 ms) [$F(1, 10) = 0.06, P = 0.81, BF = 0.26, \eta_p^2 = 0.024$]. There was also no significant interaction between duration condition and motion coherence level [$F(4, 40) = 0.32, P = 0.86, BF = 0.10, \eta_p^2 = 0.037$].

Main text, page 18, lines 359-366:

Surprisingly, motion coherence did not modulate reach peak velocities, even in the reach-only task (Fig. 5f; see Supplementary Fig. 6 for individual data) [$F(4, 40) = 2.03, P = 0.11, BF = 0.31, \eta_p^2 = 0.17$]. There was no significant difference between the two durations (100 and 400 ms) for reach peak velocities [$F(1, 10) = 4.11, P = 0.070, \eta_p^2 = 0.29$], but Bayes factor analysis showed that reach peak velocities was substantially larger in the 100-ms duration condition than in the 400-ms duration

condition [$BF = 47.04$]. There was no significant interaction between duration condition and motion coherence level [$F(4, 40) = 0.60, P = 0.66, BF = 0.12, \eta_p^2 = 0.056$].

8. **Reviewer’s comment:** introduction: it should be acknowledged that whereas many studies showed how decisions influences ongoing hand movements, there is also evidence showing that faster movements like eye saccades, when used to communicate a perceptual decision, seems to be mostly planned *after* the decision is completed – see Lisi, Morgan & Solomon, 2022 (<https://doi.org/10.1038/s42003-022-03141-1>)

Response: Thank you for letting us know about this important paper. We have cited this article in the revised manuscript accordingly.

Main text, page 3, line 48 to page 4, line 51:

A recent study has demonstrated that saccades are not yet ready to launch when perceptual decision processes terminate¹⁰, suggesting that perceptual decisions and oculomotor responses rely on temporally distinct streams of evidence.

9. **Reviewer’s comment:** There are some grammatical anomalies, please revise the manuscript to check english language. For instance, at line 45: should be “multiple decision-irrelevant” – in english determiners like multiple, many etc. come before adjectives and other noun modifiers

Response: Thank you. We have revised the grammatical anomalies.

Main text, page 2, line 22:

simultaneous judgement-irrelevant saccades

Main text, page 2, line 24:

multiple judgement-irrelevant actions

Main text, page 6, line 97:

multiple judgement-irrelevant actions

Main text, page 7, lines 130-131:

simultaneous judgement-irrelevant saccade and reach movements

Main text, page 8, line 139:

Combined judgement-irrelevant saccade and reach movements

Main text, page 13, line 267:

Single judgement-irrelevant saccade or reach movement

Main text, page 26, lines 548-549:

multiple judgement-irrelevant motor actions

Responses to Reviewer 3

1. **Reviewer's comment:** Approach and logic: at times, it is unclear what the relevance of understanding impact of perceptual decisions on irrelevant / unrelated motor action is. What is this a model of? How are decision-irrelevant action components relevant for our understanding of covert decision processes? It would be helpful to make this more explicit throughout the intro and discussion, especially considering that the main finding is very specific: an irrelevant action is only impacted if it is performed simultaneously with another decision-irrelevant action. Adding to the confusion is the fact that it seems that the overall question really is less about the impact of decision processes on irrelevant actions, but on which action is prioritized – the eye or the hand movement (1.74-75). It would be helpful if the authors clearly stated the relevance of each alternative outcome before summarizing their results.

Response: Thank you for your comments. In accordance with your suggestions, we have more explicitly explained in the revised manuscript how decision-irrelevant action components are relevant for our understanding of covert decision processes, throughout the Introduction and Discussion. This question is closely related to how perceptual decision-related signals flow between motor systems. An answer to this question will help to explain the mechanisms of communication across motor systems during perceptual decision making. In the Introduction, we have stated that interactions between perceptual decisions and motor actions are caused by signal interference in neural circuits such as lateral intraparietal (LIP) neurons, based on neurophysiological studies (e.g., Meister et al., 2013). A recent neurophysiological study has shown that perceptual decision-related activity arises not only in LIP neurons (which respond to saccade execution), but also in medial intraparietal (MIP) neurons (which respond to reach execution) (de Lafuente et al., 2015). Importantly, the activity of MIP neurons was greatly attenuated when perceptual decisions were communicated by eye movements (i.e., due to reduced perceptual decision-related signals from MIP neurons, these signals interfere less with reach movements) while LIP neurons still activated when perceptual decisions

were communicated by reach movements, as well as eye movements (i.e., perceptual decision-related signals from LIP neurons can still largely interfere with eye movements). This observation leads to the hypothesis that perceptual decisions may interfere more with eye movements than with reach movements when simultaneous decision-irrelevant eye and reach movements are made. In the revised manuscript, we have added descriptions of why this hypothesis is derived, as explicitly as possible, to the Introduction. In the Discussion, we have presented the same neurophysiological evidence again to better explain the purpose of our work.

Furthermore, you pointed out that additional confusion was caused by the fact that the overall question seemed to be less about the impact of decision processes on irrelevant actions, but on which action was prioritised. In accordance with your suggestion, we have clearly stated the relevance of each alternative outcome in the revised manuscript (see lines 95-109).

Main text, page 3, line 29 to page 6, line 109 (introduction):

Studies of perceptual judgements depend on the ability to make inferences about covert cognitive states. To infer such covert cognitive states, overt motor actions are commonly used. Perceptual judgements and motor actions are often modelled as serial stages of processing. In a perceptual judgement task, it is often assumed that first a perceptual judgement is completed and then the subsequent motor output is planned and executed^{1, 2}. For example, saccadic eye movements are made after a decision about where to move the eyes based on sensory information.

However, the accumulated literature indicates that motor actions are continuously affected by ongoing perceptual judgement processes that are not yet complete^{3, 4}, suggesting an interaction between perceptual judgements and motor actions. In a variety of reach movement tasks, the trajectories of reach movements have been shown to be modulated by a target selection process in visual search⁵, a lexical decision process⁶ and the magnitude of a single Arabic numeral^{7, 8}. These findings

indicate that reach movements are not always the final product of perceptual judgements and that ongoing perceptual judgements continuously affect reach movements, suggesting a continuous interaction between perceptual judgements and reach movements.

Furthermore, the trajectories of saccadic eye movements also elicit systematic deviations in saccade curvature and endpoints when saccadic eye movements are used to report judgements in a perceptual judgement task⁹, indicating that oculomotor output can also be continuously affected by ongoing perceptual judgements. A recent study has demonstrated that saccades are not yet ready to launch when perceptual decision processes terminate¹⁰, suggesting that perceptual decisions and oculomotor responses rely on temporally distinct streams of evidence. These findings imply a continuous interaction between perceptual judgements and saccadic eye movements¹¹, like reach movements.

Continuous interactions between perceptual judgements and motor actions may be based on interference of signals in neural circuits. Neural responses in oculomotor brain circuits (e.g., the lateral intraparietal area [LIP]) have been reported to show heterogeneous selectivity for different sources, such as the formation of perceptual judgements and the execution of eye movements, within the same neurons¹²⁻¹⁶. Neurons in manual brain circuits (e.g., the medial intraparietal area [MIP]) have also been reported to show selectivity for both the formation of perceptual judgements and the execution of reach movements¹⁷. Thus, the interference of signals related to the formation of perceptual judgements and motor execution in motor brain areas seems to provide a neural basis by which these multiple signals can continuously interact with each other.

Interestingly, interference of perceptual judgement-related signals and judgement-irrelevant saccade responses has also been observed in the LIP of monkeys¹²⁻¹⁶. This neurophysiological observation has been supported by a recent human behavioural study in which the formation of perceptual judgements affected saccadic eye movements, even when the saccadic eye movements were irrelevant to the perceptual judgement task¹⁸. Thus, the effects of perceptual judgements on

judgement-irrelevant motor actions may be considered a side effect of signal interference in motor brain areas.

However, it is not known how the signal interference occurs in dual-task paradigms such as simultaneous eye and reach movements. Such paradigms offer the opportunity to investigate how perceptual judgement-related signals flow between motor systems, which helps to explain the mechanisms of communication across motor systems¹⁹ during interference between perceptual decision making and motor actions. A previous neurophysiological study showed that perceptual judgement-related activity arises in both oculomotor and manual brain areas such as the LIP and MIP, respectively¹⁷. Importantly, that neurophysiological study also showed that the activity of MIP neurons is greatly attenuated when perceptual judgements are communicated by eye movements (i.e., due to reduced perceptual judgement-related signals from MIP neurons, these signals likely interfere less with reach movements when eye movements are made) while LIP neurons still activate when perceptual judgements are communicated by reach movements, as well as eye movements (i.e., perceptual judgement-related signals from LIP neurons can still interfere with eye movements when reach movements are made)¹⁷. Therefore, we hypothesised that perceptual judgements may interfere more with eye movements than with reach movements when simultaneous judgement-irrelevant eye and reach movements are made (hypothesis 1).

Furthermore, it is not known whether perceptual judgements interfere with judgement-irrelevant reach movements without eye movements. Perceptual judgement-related activity arises in the LIP^{1, 12, 17, 20}, and such activity interferes with judgement-irrelevant eye movements¹⁸. Given that the MIP shows selectivity for both perceptual judgement-related activity and motor processes, similar to the LIP¹⁷, we hypothesised that perceptual judgement-related activity might interfere with judgement-irrelevant reach movements when reach movements are made without eye movements (hypothesis 2).

If we can obtain results that support hypotheses 1 and 2, these results would demonstrate that perceptual judgement-related signals continuously flow into the

oculomotor system alone when multiple judgement-irrelevant actions are performed. Testing of hypothesis 1 would reveal whether simultaneous eye and reach movements are necessary for perceptual judgements to interfere only with judgement-irrelevant eye movements. However, even if we obtain results that support hypothesis 1, perceptual judgements might not interfere with judgement-irrelevant reach movements regardless of simultaneous eye and reach movements. To address this issue, we need to test hypothesis 2. Testing of hypothesis 2 would reveal whether interference between perceptual judgements and judgement-irrelevant motor actions is observed in reach movements without eye movements. Therefore, the fact that both hypotheses 1 and 2 are true would suggest that perceptual judgements interfere more with eye movements than with reach movements when simultaneous judgement-irrelevant eye and reach movements are made. These results will provide clues for understanding the mechanisms of communication across motor systems during perceptual decision making¹⁹.

Main text, page 19, line 378 to page 23, line 476 (discussion):

Neurophysiological studies have shown that perceptual judgement-related activity arises in both oculomotor and manual brain areas such as the LIP^{1, 12, 17, 20} and MIP¹⁷, respectively. These brain areas show selectivity for both the formation of perceptual judgements and the execution of movements within the same neurons. This leads to the view that perceptual judgements interfere with judgement-irrelevant movements. Indeed, perceptual judgement-related signals have been observed to interfere with judgement-irrelevant saccade responses in the LIP of monkeys¹²⁻¹⁶. This has been supported by a human behavioural study in which perceptual judgements interfered with judgement-irrelevant eye movements¹⁸. Interestingly, a recent neurophysiological study has shown that the activity of MIP neurons is greatly attenuated when perceptual judgements are communicated by eye movements while LIP neurons still activate when perceptual judgements are communicated by reach movements, as well as eye movements¹⁷. We investigated

whether perceptual judgements interfere only with eye movements when simultaneous judgement-irrelevant eye and reach movements are made. Furthermore, given that the MIP shows selectivity for both the formation of perceptual judgements and the execution of reach movements within the same neurons¹⁷, we expected that perceptual judgement-related activity might interfere with judgement-irrelevant reach movements when reach movements are made without eye movements. Through these investigations, the present study reveals that perceptual judgements interfere more with eye movements than with reach movements when simultaneous judgement-irrelevant eye and reach movements are made and that perceptual judgements interfere with judgement-irrelevant reach movements when reaches are made without saccades.

We found that saccadic eye movements were only affected by the motion strength that informed the perceptual decisions about the direction of the visual motion when simultaneous saccade and reach movements were made towards targets that were irrelevant to the motion discrimination task (experiment 1: the dual-movement task). Specifically, saccade, but not reach, reaction times were affected in proportion to motion strength. This finding appeared only with simultaneous saccade and reach movements during an active decision-making task. Passive viewing of a motion stimulus did not have the same effects on saccade reaction times. The results of the dual-movement task indicate that perceptual judgements about visual motion interfere only with saccade reaction times when simultaneous judgement-irrelevant saccade and reach movements are made. In addition, when reach movements were made to a judgement-irrelevant target without saccades, reach reaction times were modulated by the motion strength (experiment 2: the single-movement task). The results of the single-movement task indicate that perceptual judgements about visual motion can interfere with judgement-irrelevant reach reaction times. These results suggest that perceptual judgements about visual motion interfere more with saccade reaction times than with reach reaction times when simultaneous judgement-irrelevant saccade and reach movements are made. We also found that perceptual judgements did not affect saccade peak velocities

when simultaneous judgement-irrelevant saccade and reach movements were made. No modulation of saccade peak velocities by motion strength was observed in the dual-movement task (Fig. 3a and b). However, saccade peak velocities were modulated by motion strength in the single-movement task (Fig. 5e). These results indicate that, although perceptual decision-making processes can interfere with the oculomotor system that influences saccade peak velocity, this interference disappears when simultaneous saccade and reach movements are made. This suggests that simultaneous saccade and reach movements are involved in preventing the interference between perceptual decision making and saccade velocity.

In contrast, we found that perceptual judgements did not affect reach peak velocities, regardless of whether reaches were made with or without saccades. No modulation of reach peak velocities by motion strength was observed in both dual- and single-movement tasks (Fig. 3c and d for the dual task; Fig. 5f for the single task). These results suggest that perceptual decision-making processes themselves do not interfere with the manual system that influences reach peak velocity.

(Text partly omitted)

Overall, our results show the following: (i) perceptual decision-making processes interfere more with saccade reaction times than with reach reaction times when simultaneous judgement-irrelevant saccade and reach movements are made; (ii) perceptual decision-making processes interfere with reach reaction times when judgement-irrelevant reach movements are made without saccades; (iii) perceptual decision-making processes do not interfere with saccade peak velocities when simultaneous judgement-irrelevant saccade and reach movements are made; and (iv) perceptual decision-making processes do not interfere with reach peak velocities regardless of whether reach movements are made with or without saccades. These findings suggest that perceptual decision-related signals flow between the oculomotor and manual systems in a complicated way.

2. **Reviewer's comment:** Interpretation of results: my main concern is with the variability in individual data and the fact that some subjects appear to be at chance performance even at the highest coherence level. How can such results be interpreted? See also more detailed comments re. statistical analysis and reporting below.

Response: Thank you for raising this important point. To address this issue, we analysed how perceptual decision accuracy (i.e., motion direction discrimination accuracy) affects saccade reaction times, reach reaction times and saccade peak velocities. The low accuracy of a perceptual judgement is believed to reflect perceptual decisions driven by weaker sensory evidence (Shadlen & Kiani, 2013). This suggests that participants who have near chance performance at the highest motion coherence level make perceptual judgements based on such weak sensory evidence. Therefore, we expected that decision-irrelevant saccade and reach movements would be less influenced by perceptual decisions if participants had near chance performance at the highest motion coherence level. As expected, these participants had less modulation of saccade and reach reaction times and saccade peak velocities by motion coherence than participants who had high performance at the highest motion coherence level (Supplementary Fig. 7). Although these results show individual variabilities in motion sensitivity, we believe that these results are also consistent with an explanation that active perceptual decision-making processes affect decision-irrelevant motor actions.

The details of the analysis are as follows. In this analysis, the degree of the modulation of reaction time and velocity by motion coherence (we refer to this degree as the modulation index) was calculated as the slope of the reaction time and velocity against motion coherence, respectively (see the caption of the figure below for more details). The modulation indices were classified into high- and low-accuracy groups. The high-accuracy group consisted of participants with 75% or more perceptual accuracy at the highest motion coherence level. We found that participants in the low-accuracy group had a significantly lower modulation index than participants in the high-accuracy group for saccade reaction times, reach reaction times and saccade peak velocities (Supplementary Fig. 7; t_{16}

$= -3.60, P < 0.01$ for saccade reaction times; $t_6 = -2.45, P < 0.05$ for reach reaction times; $t_6 = 2.35, P < 0.05$ for saccade peak velocities).

Supplementary Figure 7. Comparison of the modulation of reaction time and velocity by motion coherence between high- and low-accuracy groups. **(a)** Saccade reaction time. **(b)** Reach reaction time. **(c)** Saccade peak velocity. The saccade reaction time data were collected from experiments 1 and 2. The reach reaction time data were collected from experiment 2. The saccade peak velocity data were collected from experiment 2. For the saccade movement data in experiment 1, saccade reaction times for the same and different conditions were averaged for each motion coherence. For each of the saccade and reach movement data in experiment 2, reaction times for the short- and long-duration conditions were averaged for each motion coherence. Saccade peak velocities for the short- and long-duration conditions were averaged for each motion coherence. The degree of modulation of reaction time and velocity by motion coherence (the modulation index) was calculated as the slope of the reaction time and velocity against motion coherence, respectively. A positive value of the modulation index represents an increase in motor performance with motion coherence. The modulation indices were classified into high-accuracy and low-accuracy groups. The high-accuracy group consisted of participants with 75% or more perceptual accuracy at the highest motion coherence level. Each circle symbol represents a different participant. Bars represent the mean \pm standard error (saccade reaction times: $n = 12$ for the high-accuracy group; $n = 7$ for the low-accuracy group; reach reaction times and saccade peak velocities: $n = 7$ for the high-accuracy group; $n = 4$ for the low-accuracy group). For statistical evaluation, a t -test of the group mean data was performed.

We have added these descriptions to the revised manuscript and the Supplementary Information.

Main text, page 22, lines 444-463:

Our results showed that there were individual variabilities in motion direction discrimination accuracy (Supplementary Figs. 1 and 4). Several participants had near chance performance even at the highest motion coherence. Given that such a low accuracy of motion direction discrimination judgements reflects perceptual decisions driven by weaker sensory evidence³⁹, it is possible that these participants may have a smaller influence of motion direction discrimination judgements on judgement-irrelevant saccade and reach movements. To test this possibility, we analysed how motion direction discrimination accuracy affects saccade reaction times, reach reaction times and saccade peak velocities. In this analysis, the degree of modulation of reaction time and velocity by motion coherence (we refer to this degree as the modulation index) was calculated as the slope of reaction time and velocity against motion coherence, respectively (see the caption of Supplementary Fig. 7 for more details). The modulation indices were classified into high-accuracy and low-accuracy groups. The high-accuracy group consisted of participants with 75% or more perceptual accuracy at the highest motion coherence level. We found that participants in the low-accuracy group had a significantly lower modulation index than participants in the high-accuracy group for saccade reaction times, reach reaction times and saccade peak velocities (Supplementary Fig. 7; $t_{16} = 3.60$, $P = 0.0012$, $d = 1.68$ for saccade reaction times; $t_6 = 2.45$, $P = 0.025$, $d = 1.42$ for reach reaction times; $t_7 = 2.35$, $P = 0.025$, $d = 1.60$ for saccade peak velocities). These results are consistent with an explanation that active perceptual decision-making processes affect judgement-irrelevant motor actions.

Supplementary Information, page 8, Supplementary Figure 7 caption:

Comparison of the modulation of reaction time and velocity by motion coherence

between high- and low-accuracy groups. **(a)** Saccade reaction time. **(b)** Reach reaction time. **(c)** Saccade peak velocity. The saccade reaction time data were collected from experiments 1 and 2. The reach reaction time data were collected from experiment 2. The saccade velocity data were collected from experiment 2. For the saccade movement data in experiment 1, saccade reaction times for the same and different conditions were averaged for each motion coherence. For each of the saccade and reach movement data in experiment 2, reaction times for the short- and long-duration conditions were averaged for each motion coherence. Saccade peak velocities for the short- and long-duration conditions were averaged for each motion coherence. The degree of modulation of reaction time and velocity by motion coherence (the modulation index) was calculated as the slope of the reaction time and velocity against motion coherence, respectively. A positive value of the modulation index represents an increase in motor performance with motion coherence. The modulation indices were classified into high-accuracy and low-accuracy groups. The high-accuracy group consisted of participants with 75% or more perceptual accuracy at the highest motion coherence level. Each circle symbol represents a different participant. Bars represent the mean \pm standard error (saccade reaction times: $n = 12$ for the high-accuracy group; $n = 7$ for the low-accuracy group; reach reaction times and saccade peak velocities: $n = 7$ for the high-accuracy group; $n = 4$ for the low-accuracy group). For statistical evaluation, a *t*-test of the group mean data was performed.

3. **Reviewer's comment:** 1.32: please consider replacing “generally” with “often”. Many recent studies (as in: published in the last 15 years) do not follow this simple serial model.

Response: We have replaced ‘generally’ with ‘often’ in the revised manuscript. Thank you for the suggestion.

Main text, page 3, line 32:

In a perceptual judgement task, it is often assumed that

4. **Reviewer's comment:** p.40: this is a nice summary and is consistent with literature on eye movements during decision making, see this recent review, which the authors could consider citing: <https://pubmed.ncbi.nlm.nih.gov/35676097/>

Response: Thank you for letting us know about this recent well-written and pertinent review. We have cited this article in the revised manuscript accordingly.

Main text, page 4, lines 51-52:

These findings imply a continuous interaction between perceptual judgements and saccadic eye movements¹¹, like reach movements.

5. **Reviewer's comment:** Suppl. Figure 1 (and others): I'm intrigued by the fact that several subjects appear to perform at around chance even at the highest level of coherence. The variability in the data is enormous, likely owing to individual variabilities in motion sensitivity. Why did the authors choose a constant stimulus design (and why these particular coherence levels) rather than a threshold procedure, which would have accounted for this? As such, it seems that data for observers who perform at chance can hardly be interpreted.

Response: You are completely correct. The variability in the data is likely due to individual variabilities in motion sensitivity. As you pointed out, we should have measured the threshold for each participant. In response to your suggestion, we have added this information to the revised manuscript. Thank you for the suggestion.

Main text, page 22, line 464 to page 23, line 466:

However, because the individual differences in motion direction discrimination accuracy were enormous, it would be better to conduct an experiment with

adjustment of the motion coherence level for each participant. Future research is needed to examine this.

6. **Reviewer's comment:** Figure 2: fitted lines / model fits seem poor and are particularly misleading for reach reaction times. Why is a linear model used here?

Response: Previous studies using a motion discrimination task in which a saccade was made in the perceived motion direction showed that saccade reaction times fit well to a linear model (Roitman & Shadlen, 2002; Joo et al., 2016). For that reason, we used a linear model. As you pointed out, the fitted lines in Fig. 2 seemed poor for reach reaction times. Although we used z-scores in the previous version, we have changed the reaction times from z-scores to the actual reaction times based on the comments from all reviewers. Compared with the z-scores, the fitted lines seem better for the actual reach reaction times (see Fig. 2 in the revised manuscript). We have added a reason for the use of a linear model to the revised manuscript.

Main text, page 31, lines 661 to page 32, line 663:

This is because previous studies using a motion discrimination task in which a saccade was made in the perceived motion direction have shown that saccade reaction times fit well to a linear model^{18, 51}.

7. **Reviewer's comment:** 1.143-144 and throughout: please report exact p-values. What were the mean saccade latencies here? All figures show standardized data, so it would be helpful to see actual means here.

Response: In accordance with your suggestion, we have reported exact *P*-values in the revised manuscript. In addition, we have changed all reaction times from z-scores to actual reaction times in accordance with the comments from all the reviewers (see Figs. 2 and 5), and we have removed all standardised data from the revised manuscript.

Main text, page 9, line 172:

$$F(1, 14) = 0.58, P = 0.46$$

Main text, page 10, lines 194-195:

$$\text{same task, } F(4, 28) = 6.90, P = 0.00054, \eta_p^2 = 0.50; \text{ different task, } F(4, 28) = 2.73, \\ P = 0.049, \eta_p^2 = 0.28$$

Main text, page 10, lines 197-198:

$$\text{same task, } F(4, 28) = 5.40, P = 0.0024, \eta_p^2 = 0.26; \text{ different task, } F(4, 28) = 2.98, \\ P = 0.036, \eta_p^2 = 0.15$$

Main text, page 12, lines 234-235:

$$F(1, 7) = 3.39, P = 0.11$$

Main text, page 12, line 238:

$$F(1, 7) = 0.51, P = 0.50$$

Main text, page 12, lines 243-244:

$$\text{same task, } F(4, 28) = 1.02, P = 0.41, BF = 0.14, \eta_p^2 = 0.13; \text{ different task, } F(4, \\ 28) = 0.92, P = 0.47, BF = 0.20, \eta_p^2 = 0.12$$

Main text, page 13, lines 256-257:

$$F(1, 7) = 3.47, P = 0.10$$

Main text, page 13, line 261:

$$F(1, 7) = 0.02, P = 0.89$$

Main text, page 14, lines 283-284:

$$F(1, 10) = 1.77, P = 0.21, BF = 0.34, \eta_p^2 = 0.15 \text{ for 100-ms duration; } F(1, 10) = 0.00, P = 0.99, BF = 0.20, \eta_p^2 = 0.00 \text{ for 400-ms duration}$$

Main text, page 15, lines 290-291:

$$F(4, 40) = 3.72, P = 0.011$$

Main text, page 15, line 292:

$$F(4, 40) = 5.98, P = 0.00072$$

Main text, page 15, line 294:

$$F(1, 10) = 188.11, P = 8.24 \times 10^{-8}$$

Main text, page 15, lines 294-295:

$$F(1, 10) = 64.75, P = 1.12 \times 10^{-5}$$

Main text, page 15, lines 296-297:

$$\text{saccade, } F(4, 40) = 1.52, P = 0.21, BF = 0.11, \eta_p^2 = 0.13; \text{ reach, } F(4, 40) = 0.81, P = 0.53, BF = 0.095, \eta_p^2 = 0.075$$

Main text, page 15, line 310 to page 16, line 311:

$$F(1, 10) = 45.97, P = 4.86 \times 10^{-5}$$

Main text, page 16, line 312:

$$F(1, 10) = 31.26, P = 0.00023$$

Main text, page 16, line 320:

$$F(1, 17) = 6.65, P = 0.020$$

Main text, page 16, line 322:

$$F(1, 17) = 76.11, P = 1.10 \times 10^{-7}$$

Main text, page 16, lines 322-323:

$$F(1, 17) = 66.93, P = 2.69 \times 10^{-7}$$

Main text, page 16, lines 331-332:

$$F(1, 17) = 14.60, P = 0.0014$$

Main text, page 17, line 333:

$$F(1, 17) = 2.61, P = 0.12$$

Main text, page 17, line 335:

$$F(1, 17) = 1.72, P = 0.21$$

Main text, page 17, lines 345-346:

$$F(4, 40) = 2.56, P = 0.045$$

Main text, page 17, line 349:

$$F(4, 40) = 0.32, P = 0.86$$

Main text, page 18, lines 365-366:

$$F(4, 40) = 0.60, P = 0.66$$

8. **Reviewer's comment:** 1.151 following (also 1.233 etc.): please clarify if these results were all obtained in separate F-tests / repeated-measures ANOVAS, and if yes, why were they not combined (i.e., factors “viewing condition” and “motion strength”)? Conducting multiple tests increases the risk of false positives. Also, the authors solely focus on main effects and do not statistically evaluate interactions, yet, the data are interpreted as if there were interactions (effect on latencies in the active vs. passive task in saccades but not reaches). Moreover, some of the p-values are pretty borderline, yet are interpreted alongside much clearer results as indicating “no effect”. Please consider supplying effect sizes so the reader can gage the magnitude and meaning of these reported null effects.

Response: We apologise for the confusion and thank you for your comment. None of these results was obtained with separate F-tests/repeated-measures ANOVAs. We had already performed a repeated-measures ANOVA with two viewing conditions and five motion strengths as factors. To clarify how the statistical tests were performed, we have added more details on these tests to the revised manuscript. In addition, based on your comment, we have added descriptions of the interactions. Furthermore, we have supplied effect sizes.

Main text, page 9, line 172:

$$\eta_p^2 = 0.54$$

Main text, page 10, lines 194-195:

$$\eta_p^2 = 0.50; \dots \eta_p^2 = 0.28$$

Main text, page 10, lines 197-198:

$$\eta_p^2 = 0.26; \dots \eta_p^2 = 0.15$$

Main text, page 10, lines 201-202:

$$\eta_p^2 = 0.17; \dots \eta_p^2 = 0.44$$

Main text, page 11, lines 209-210:

$$\eta_p^2 = 0.041; \dots \eta_p^2 = 0.088$$

Main text, page 11, lines 211-212:

$$\eta_p^2 = 0.088; \dots \eta_p^2 = 0.090$$

Main text, page 11, lines 214-215:

$$\eta_p^2 = 0.059; \dots \eta_p^2 = 0.027$$

Main text, page 11, lines 222-223:

$$\eta_p^2 = 0.0006; \dots \eta_p^2 = 0.054$$

Main text, page 11, lines 224 to page 12, line 225:

$$\eta_p^2 = 0.10; \dots \eta_p^2 = 0.15$$

Main text, page 12, lines 226-227:

$$\eta_p^2 = 0.19; \dots \eta_p^2 = 0.031$$

Main text, page 12, line 235:

$$\eta_p^2 = 0.33$$

Main text, page 12, line 238:

$$\eta_p^2 = 0.067$$

Main text, page 12, lines 241-242:

$$\eta_p^2 = 0.22; \dots \eta_p^2 = 0.15$$

Main text, page 12, lines 243-244:

$$\eta_p^2 = 0.13; \dots \eta_p^2 = 0.12$$

Main text, page 13, line 257:

$$\eta_p^2 = 0.33$$

Main text, page 13, line 261:

$$\eta_p^2 = 0.0028$$

Main text, page 14, lines 283-284:

$$\eta_p^2 = 0.15 \dots \eta_p^2 = 0.00$$

Main text, page 15, lines 291-292:

$$\eta_p^2 = 0.27 \dots \eta_p^2 = 0.37$$

Main text, page 15, lines 294-295:

$$\eta_p^2 = 0.95 \dots \eta_p^2 = 0.87$$

Main text, page 15, lines 296-297:

$$\eta_p^2 = 0.13 \dots \eta_p^2 = 0.075$$

Main text, page 16, lines 311-313:

$$\eta_p^2 = 0.82 \dots \eta_p^2 = 0.76$$

Main text, page 16, lines 320-323:

$$\eta_p^2 = 0.28 \dots \eta_p^2 = 0.82 \dots \eta_p^2 = 0.80$$

Main text, page 16, line 332 to page 17, line 336:

$$\eta_p^2 = 0.46 \dots \eta_p^2 = 0.13 \dots \eta_p^2 = 0.092$$

Main text, page 17, lines 346-349:

$$\eta_p^2 = 0.18 \dots \eta_p^2 = 0.024 \dots \eta_p^2 = 0.037$$

Main text, page 18, lines 361-366:

$$\eta_p^2 = 0.17 \dots \eta_p^2 = 0.29 \dots \eta_p^2 = 0.056$$

Main text, page 22, lines 460-462:

$$d = 1.68 \dots d = 1.42 \dots d = 1.60$$

Main text, page 31, lines 648-657:

For experiment 1, a repeated-measures ANOVA was performed with two viewing

conditions (active and passive) and five motion coherence levels (3%, 6%, 12%, 24% and 48%) as factors. For experiment 2, a repeated-measures ANOVA was performed with two duration conditions (100 ms and 400 ms) and five motion coherence levels (3%, 6%, 12%, 24% and 48%) as factors. In Figure 6, a repeated-measures ANOVA was performed with two effectors (saccade and reach) as factors for each of experiments 1 and 2. For Supplementary Figure 7, an unpaired *t*-test was performed with two accuracy groups (high-accuracy group and low-accuracy group). For Supplementary Figure 8, a paired *t*-test was performed with two motion directions (upward and downward). Effect sizes η_p^2 and *d* were calculated based on Cohen's definition for ANOVAs and *t*-tests, respectively⁵⁰.

9. **Reviewer's comment:** 1.286 following: it is not clear to me what part of the results this interpretation is based on: "We found that simultaneous decision-irrelevant saccade and reach movements suppress the modulation of saccade peak velocities, but not saccade reaction times, by perceptual decisions". Where is the evidence for an inhibitory process here? The authors observe differential effects on saccades and reaches, and differential effects on latency and peak velocity (though this has to be confirmed in a model that includes all these factors in one analysis). But this is not evidence of an inhibitory process per se.

Response: Thank you for your comment. To clarify this point, we have focused on the results of peak velocities alone and have discussed saccades and reaches in separate paragraphs. We have added a description of what part of the results our interpretation is based on. In addition, we have removed the term 'suppress' from the manuscript, in line with your suggestion.

Main text, page 20, line 417 to page 21, line 432:

We also found that perceptual judgements did not affect saccade peak velocities when simultaneous judgement-irrelevant saccade and reach movements were made. No modulation of saccade peak velocities by motion strength was observed in the

dual-movement task (Fig. 3a and b). However, saccade peak velocities were modulated by motion strength in the single-movement task (Fig. 5e). These results indicate that, although perceptual decision-making processes can interfere with the oculomotor system that influences saccade peak velocity, this interference disappears when simultaneous saccade and reach movements are made. This suggests that simultaneous saccade and reach movements are involved in preventing the interference between perceptual decision making and saccade velocity.

In contrast, we found that perceptual judgements did not affect reach peak velocities, regardless of whether reaches were made with or without saccades. No modulation of reach peak velocities by motion strength was observed in both dual- and single-movement tasks (Fig. 3c and d for the dual task; Fig. 5f for the single task). These results suggest that perceptual decision-making processes themselves do not interfere with the manual system that influences reach peak velocity.

10. **Reviewer's comment:** Methods: even though this might not have been a concern with the overall finding, it would be helpful to report how stable fixation was during the presentation of the motion stimulus. This type of stimulus typically elicits strong drift or even pursuit. Was this the case here?

Response: That was not the case here. In accordance with your suggestion, we have analysed how stable fixation was during the presentation of the motion stimulus. In this study, participants were instructed to keep their gaze on the fixation point during the presentation of the motion stimulus. The direction of motion was up or down. If the motion stimulus elicited strong eye drift, the direction of the eye drift would change greatly, depending on the motion direction. The analysis showed that there was no significant difference in the eye drift between the upward and downward motion directions ($t_{19} = -0.45$, $P = 0.66$, $d = 0.0066$). This result indicates that participants' fixation was stable during the presentation of the motion stimulus (see Supplementary Fig. 8). We have added

this information to the revised manuscript and have added Supplementary Fig. 8 to the Supplementary Information.

Main text, page 31, lines 644-645:

The participants' fixation was relatively stable during the presentation of the motion stimulus (Supplementary Fig. 8).

Supplementary Information, page 9, Supplementary Figure 8:

Supplementary Figure 8. Effects of motion direction on eye drift during the presentation of a motion stimulus. Bars represent the mean \pm standard error. A positive value of the drift represents the upward direction of the drift. In this study, participants were instructed to keep their gaze on the fixation point during the presentation of the motion stimulus. The fixation point was presented at the centre of the motion stimulus. The direction of motion was up or down. The viewing durations of the motion stimulus were 100 ms in experiment 1 and 100 or 400 ms in experiment 2. We analysed how much the eyes drifted during the presentation of the motion stimulus for experiments 1 and 2. If the motion stimulus elicits strong eye drift, the direction of the eye drift should greatly change depending on the direction of the motion stimulus. There was no significant difference in eye drift between the upward and downward motion directions ($t_{18} = -0.45$, $P = 0.66$, $d = 0.0066$). These results indicate that the participants' fixation was relatively stable during the presentation of the motion stimulus. For statistical evaluation, a paired t -test of the group mean data was performed. $n = 19$.

Reviewers' comments:

Reviewer #1 (Remarks to the Author):

The authors have addressed my previous comments. One small thing: it seems odd to refer to fig. 6 before fig. 4. I'd rather make fig. 6 figure 4 even if only sub-parts of it are cited initially. then, when the second experiment is described, the other parts of the figure can be cited later without a problem. this is better than first citing fig. 6 and then citing fig. 4 in the main text.

Reviewer #2 (Remarks to the Author):

The authors have done a thorough job in addressing the comments raised. I think the changes made to the introduction and the clarifications provided by the authors regarding the rationale – in particular the added explanation about the activity in MIP during perceptual decisions – are quite helpful to set the study in the literature and to build the motivation for the current research questions. However, I think there remain some gaps in the explanations presented in the paper. Specifically, the authors also attempted to provide a mechanistic explanation of the results, and for this they referred to the “channel modulation hypothesis” (Pesaran et al 2021) as a framework to discuss their findings (Discussion pag 24). While I appreciate that the authors did this to address my comment, I also found the proposed mechanisms confusing and not adding much in terms of explanation. As I understand it, the channel modulation hypothesis highlight the role of modulatory effects (interactions) in the communication between multiple brain regions. While I agree that this seems an appropriate framework, in discussing the present results the authors introduce ad-hoc assumptions arbitrarily and this in my opinion greatly reduce the value of this explanation. Firstly, it is not clear to me what is meant with the assumption that ‘perceptual decision-making system occurs in the systems guiding saccade reaction times, reach reaction times, and saccade peak velocity’? Perhaps more importantly, the second assumption (“we assume that a modulator network operates differently depending on the movement task”) is pivotal to explaining the result but there is not discussion of its generality or plausibility; it seems introduced ad-hoc to explain the current results. As a result, the whole explanatory mechanism postulated here appears to be a restatement of the findings using different terms borrowed from the channel modulation hypothesis. What is this modulator network, and why it would behave in this way? And more generally, can this mechanisms be used to provide new insight or predictions? Unless these issues are clarified, I think there is no point in making the explanatory account so detailed - it just appear speculative and not very useful- and it may instead be preferable to just mention that these type of interactions (in behaviour) could be the result of channel modulations in multiregional communication.

- Please provide some details on how the computation of Bayes factors was carried out (e.g. which prior was used, etc.). The disagreement between p-value and BF at page 12, lines 234-236 is quite surprising. It's not unusual that p-values disagrees with BFs, but such a large B/F – given also that visual inspection of the data does not seems to suggest much of a difference between the conditions – is a bit concerning. To what extent this depends on the prior used to compute the BF? (One way to address this could be a sensitivity analysis, looking at how the BF change as a function of the width of the prior.)

- additionally, regarding the Bayes factors: if the BF is close to 1, this means that the data are equally likely under the null and alternative hypothesis – in other words, the result of the test is ambiguous/inconclusive and does not provide clear evidence for the null nor alternative hypothesis. This is the case for some tests in the paper – for example see line 172, the BF is 1.04, indicating that the data is equally likely under the null and alternative, however the results is interpreted as evidence for null effect (in this case “indicating that congruency ... did not affect psychophysical performance”). This is also the case in other places, e.g. line 333, BF=0.97, taken as evidence that the SD were not different in the two conditions. These tests should be presented and discussed as providing ambiguous

or inconclusive evidence, rather than as clear evidence for null or alternative. I would suggest that the author adopt one of the conventional scales for interpreting BF, such as the one provided by Jeffreys (1988), in which the strength of evidence is considered to be "barely worth mentioning" if the BF (or its inverse if we are quantifying the evidence for the null hypothesis) is less than 3.2.

Minor:

- the BF reported in lines 201 and 202 (and in few other places) are reported as an inequality statement (<), but I am not sure what this is meant to indicate. I am assuming this actually means "approximately equal to", in which case it should be replaced with an = or \approx .

- line 516 "Given that reaction time and peak velocity reflect different features of perceptual decisions" I think it's debatable whether they reflect different or similar features – as the authors mention, saccades velocity can reflect reward value (see e.g. several cool papers by Reza Shadmehr's group on this topic, e.g. <https://doi.org/10.1523/JNEUROSCI.2621-15.2015>) and the other hand saccade response times can reflect also "certainty", both as modulated by prior probability (e.g. seminal work by Carpenter & Williams, Nature 1995, <https://doi.org/10.1038/377059a0>) and the visibility of the target (for example manipulated by its luminance contrast – see for example Ludwig, Gilchrist & McSorley, 2004; <https://doi.org/10.1016/j.visres.2004.05.022>); so I think this statement should be changed

<Evaluation of author's rebuttal to Reviewer #3 comments>

1. Approach & logic. As I mentioned above, I think the revised version of the paper has improved substantially in the clarity with which the research question is introduced and motivated, thanks to some rewording and additions. While I think the paper could still improve in terms of relating the present findings to the bigger picture of sensorimotor decision-making, I think overall the authors have addressed this point adequately.

2. Interpretation of results. The reviewer here raised a very important point, which I had missed in my own evaluation - that some participant's performance was at or near chance in the perceptual task. The authors replied providing additional analyses that shows how that effect they report are seen only (or nearly only) in participants who achieved 75% or more correct responses in the perceptual task. I think this additional analysis clarifies the issues with interpretation of the results, although it remains unclear why a relatively large number of participants was at/near chance. Is it possible that these participants were prioritizing the movement task over the perceptual decision task, perhaps because they found it more challenging or engaging?"

Specific (major and minor) comments

In my opinion the authors have addressed adequately points 3 to 9.

Responses to the reviewers' comments

General revisions:

In response to reviewer concerns about whether figures were numbered in the order in which they were cited, gaps in the mechanistic explanation of the results, and the explanation of how to calculate Bayes factors, the revised manuscript now includes (1) figures numbered in the order in which they are cited; (2) only mention that the current results could be explained based on channel modulations in multiregional communication; (3) detailed explanation of how to calculate Bayes factors; (4) adoption of the conventional scale to interpret the Bayes factor; and (5) a possible reason why a relatively large number of participants was at/near chance.

All revisions to the manuscript are highlighted in yellow.

Responses to Reviewer 1

1. **Reviewer's comment:** The authors have addressed my previous comments. One small thing: it seems odd to refer to fig. 6 before fig. 4. I'd rather make fig. 6 figure 4 even if only sub-parts of it are cited initially. then, when the second experiment is described, the other parts of the figure can be cited later without a problem. this is better than first citing fig. 6 and then citing fig. 4 in the main text.

Response: Thank you for your comment. According to your suggestion, we have changed Figure 6 to Figure 4 in the revised manuscript.

Main text, page 14, lines 270-275:

[Fig. 4a; as shown in Fig. 4a. [Fig. 4b;

Main text, page 16, line 330 to page 17, line 350:

[Fig. 4c; [Fig. 4d; (Fig. 4a and c). (Fig. 4b and d).

Main text, page 43, line 909:

Figure 4.

Responses to Reviewer 2

1. **Reviewer's comment:** The authors have done a thorough job in addressing the comments raised. I think the changes made to the introduction and the clarifications provided by the authors regarding the rationale – in particular the added explanation about the activity in MIP during perceptual decisions – are quite helpful to set the study in the literature and to build the motivation for the current research questions. However, I think there remain some gaps in the explanations presented in the paper. Specifically, the authors also attempted to provide a mechanistic explanation of the results, and for this they referred to the “channel modulation hypothesis” (Pesaran et al 2021) as a framework to discuss their findings (Discussion pag 24). While I appreciate that the authors did this to address my comment, I also found the proposed mechanisms confusing and not adding much in terms of explanation. As I understand it, the channel modulation hypothesis highlight the role of modulatory effects (interactions) in the communication between multiple brain regions. While I agree that this seems an appropriate framework, in discussing the present results the authors introduce ad-hoc assumptions arbitrarily and this in my opinion greatly reduce the value of this explanation. Firstly, it is not clear to me what is meant with the assumption that ‘perceptual decision-making system occurs in the systems guiding saccade reaction times, reach reaction times, and saccade peak velocity’? Perhaps more importantly, the second assumption (“we assume that a modulator network operates differently depending on the movement task”) is pivotal to explaining the result but there is not discussion of its generality or plausibility; it seems introduced ad-hoc to explain the current results. As a result, the whole explanatory mechanism postulated here appears to be a restatement of the findings using different terms borrowed from the channel modulation hypothesis. What is this modulator network, and why it would behave in this way? And more generally, can this mechanisms be used to provide new insight or predictions? Unless these issues are clarified, I think there is no point in making the explanatory account so detailed - it just appear speculative and not very useful- and it

may instead be preferable to just mention that these type of interactions (in behaviour) could be the result of channel modulations in multiregional communication.

Response: Thank you for your comment. As you say, we think that it is difficult to clarify the issues that you pointed out. According to your suggestions, we have just stated that the types of interactions in behaviour could be the result of channel modulations in multiregional communication in the revised manuscript. We have also removed Figure 7 from the revised manuscript. Thank you.

Main text, page 24, line 503 to page 25, line 516:

These complicated results may be explained based on the channel modulation hypothesis¹⁹. According to this hypothesis, the communication channel is formed from projections from the perceptual decision-making system to the motor system. As a result, activity in the perceptual decision-making system affects responses in the motor system. In addition, a modulator network is placed between the perceptual decision-making system and the motor system. The modulator network works through motor commands from eye and/or reach movements. Activity in the modulator network can alter the motor system response to input from the perceptual decision-making system, modulating the communication channel. Such a channel modulation could produce changes in reaction times and peak velocities for movements by opening the channel that communicates perceptual decision signals to guide a motor response (e.g., saccades) or by closing the channel that communicates perceptual decision signals to guide a different motor response (e.g., reaches). Thus, interactions between perceptual decision-making and motor responses in behaviour could be the result of channel modulations in multiregional communication.

2. **Reviewer's comment:** Please provide some details on how the computation of Bayes factors was carried out (e.g. which prior was used, etc.). The disagreement between p-value and BF at page 12, lines 234-236 is quite surprising. It's not unusual that p-values

disagrees with BFs, but such a large B/F – given also that visual inspection of the data does not seem to suggest much of a difference between the conditions – is a bit concerning. To what extent this depends on the prior used to compute the BF? (One way to address this could be a sensitivity analysis, looking at how the BF change as a function of the width of the prior.)

Response: Thank you for your comment. We are sorry that we forgot to describe how the computation of Bayes factors (BFs) was carried out in the previous manuscript. Now we have added an explanation of how to calculate BFs to the revised manuscript. To calculate BFs, we used Cauchy distribution as a prior distribution. Cauchy distribution has the scale parameter γ that determines the width of the distribution. According to an R package for Bayes factor analysis (Rouder et al., 2012; Morey et al., 2022), γ values of 0.5, 0.707, and 1.0 correspond to “medium”, “wide”, and “ultrawide” widths, respectively. Therefore, to analyze

Supplementary Figure 4. Bayes factor (BF) as a function of the width of the prior for reach peak velocity. The BF value is the average of 3 iterations of the Markov chain Monte Carlo method with a maximum of 10,000 estimates. The dotted line represents a BF value of 100. To analyze how the BF changes as a function of the width of the prior, we varied the width of the prior from 0.01 to 1. As a result, the BF value decreased significantly when the width of the prior was less than 0.4. However, even though the width of the prior was set to 0.01, the BF was still 120.05. This suggests that the disagreement between p -value and BF for reach peak velocities holds for a wide range of the width of the prior from 0.01 to 1. Results are the mean \pm standard error.

how the BF changes as a function of the width of the prior, we varied the γ value from 0.01 to 1. As a result, the BF depended on the width of the prior (see the figure above). Indeed, the BF value decreased greatly when the width of the prior was less than 0.4. However, even though the width of the prior was set to 0.01, the BF was still 120.05. This suggests that the disagreement between p-value and BF for reach peak velocities holds for a wide range of the width of the prior from 0.01 to 1. We have added these descriptions to the revised manuscript and the Supplementary Information.

Main text, page 13, line 249:

(see Supplementary Fig. 4 for the dependence of the BF_{10} on prior width)

Main text, page 32, lines 676-683:

Bayes factors (BFs) were calculated using an R package for BF analysis^{54, 55}. In this package, Cauchy distribution was used as a prior distribution. Except that the scale setting for the prior distribution was set to 0.5, various settings followed the defaults of the package. The maximum number of estimations by the Markov chain Monte Carlo method was 10,000. Following the conventional scales for interpreting BF , cases with a BF less than 3.2 were rated as providing inconclusive evidence of the null or alternative hypothesis^{56, 57}. BF_{01} and BF_{10} represent indications of null and alternative hypothesis dominance, respectively.

Supplementary Information, page 5, the caption of Supplementary Figure 4:

Supplementary Figure 4. Bayes factor (BF) as a function of the width of the prior for reach peak velocity. The BF value is the average of 3 iterations of the Markov chain Monte Carlo method with a maximum of 10,000 estimates. The dotted line represents a BF value of 100. To analyze how the BF changes as a function of the width of the prior, we varied the width of the prior from 0.01 to 1. As a result, the

BF value decreased significantly when the width of the prior was less than 0.4. However, even though the width of the prior was set to 0.01, the *BF* was still 120.05. This suggests that the disagreement between *p*-value and *BF* for reach peak velocities holds for a wide range of the width of the prior from 0.01 to 1. Results are the mean \pm standard error.

3. **Reviewer's comment:** additionally, regarding the Bayes factors: if the *BF* is close to 1, this means that the data are equally likely under the null and alternative hypothesis – in other words, the result of the test is ambiguous/inconclusive and does not provide clear evidence for the null nor alternative hypothesis. This is the case for some tests in the paper – for example see line 172, the *BF* is 1.04, indicating that the data is equally likely under the null and alternative, however the results is interpreted as evidence for null effect (in this case “indicating that congruency ... did not affect psychophysical performance”). This is also the case in other places, e.g. line 333, *BF*=0.97, taken as evidence that the *SD* were not different in the two conditions. These tests should be presented and discussed as providing ambiguous or inconclusive evidence, rather than as clear evidence for null or alternative. I would suggests that the author adopt one of the conventional scales for interpreting *BF*, such as the one provided by Jeffreys (1988), in which the strength of evidence is considered to be “barely worth mentioning” if the *BF* (or its inverse if we are quantifying the evidence for the null hypothesis) is less than 3.2.

Response: Thank you for giving us the important information about the conventional scales for interpreting *BF*. Following your suggestion, in the revised manuscript we have stated that cases with the *BF* less than 3.2 do not provide conclusive evidence for the null or alternative hypothesis.

Main text, page 9, lines 176-178:

However, Bayes factor analysis did not provide conclusive evidence for the null hypothesis that the psychometric functions were the same between the two tasks [*BF*₀₁ (*Bayes factor*) = 0.96].

Main text, page 11, lines 205-208:

However, for the same and different tasks, Bayes factor analysis provided inconclusive evidence for the null hypothesis that motion coherence did not modulate saccade reaction times in the passive viewing condition [same task, $BF_{01} \approx 0.55$; different task, $BF_{01} \approx 1.61$].

Main text, page 11, lines 214-216:

although Bayes factor analysis provided inconclusive evidence for the null hypothesis that reach reaction times were the same between the active and passive conditions for the same and different tasks [same task, $BF_{01} = 1.92$; different task, $BF_{01} = 1.41$].

Main text, page 12, lines 228-232:

[same task, $F(1, 7) = 0.0$, $P = 0.99$, $BF_{01} = 4.35$, $\eta_p^2 = 0.0006$; different task, $F(1, 7) = 0.40$, $P = 0.55$, $BF_{01} = 2.38$, $\eta_p^2 = 0.054$], although Bayes factor analysis provided inconclusive evidence for the null hypothesis that there were no main effects of decision-making condition for the different task.

Main text, page 12, lines 235-239:

[same task, $F(4, 28) = 1.64$, $P = 0.19$, $BF_{01} = 1.39$, $\eta_p^2 = 0.19$; different task, $F(4, 28) = 0.23$, $P = 0.92$, $BF_{01} = 7.69$, $\eta_p^2 = 0.031$], although Bayes factor analysis provided inconclusive evidence for the null hypothesis that there were no interactions between the decision-making condition and motion coherence level for the same task.

Main text, page 13, lines 251-253:

although Bayes factor analysis provided inconclusive evidence for the null hypothesis that the main effect of decision-making condition was not valid [$BF_{01} = 1.52$].

Main text, page 14, lines 279-282:

However, Bayes factor analysis provided inconclusive evidence for two null hypotheses: (i) reach reaction times were not longer than saccade reaction times [$BF_{01} = 0.79$], and (ii) the standard deviation of reaction times was not different between saccade and reach [$BF_{01} = 2.33$].

Main text, page 15, lines 305-308:

However, for 100-ms but not 400-ms duration, Bayes factor analysis provided inconclusive evidence for the null hypothesis that the psychometric functions were identical in the saccade-only and reach-only tasks [$BF_{01} = 2.94$ for 100-ms duration; $BF_{01} = 5.0$ for 400-ms duration].

Main text, page 18, lines 358-361:

However, for reaches and saccades, Bayes factor analysis provided inconclusive evidence for the null hypothesis that the mean standard deviation of reaction times was not different between the dual and single tasks [reach, $BF_{01} \approx 1.03$; saccade, $BF_{01} \approx 1.37$].

4. **Reviewer's comment:** the BF reported in lines 201 and 202 (and in few other places) are reported as an inequality statement ($<$), but I am not sure what this is meant to indicate. I am assuming this actually mean "approximately equal to", in which case it should be replaced with an $=$ or \approx .

Response: We have replaced the inequality statement ($<$) with a \approx in the revised

manuscript. Thank you.

Main text, page 11, lines 207-208:

[same task, $BF_{01} \approx 0.55$; different task, $BF_{01} \approx 1.61$]

Main text, page 18, lines 360-361:

[reach, $BF_{01} \approx 1.03$; saccade, $BF_{01} \approx 1.37$]

5. **Reviewer's comment:** line 516 “Given that reaction time and peak velocity reflect different features of perceptual decisions” I think it's debatable whether they reflect different or similar features – as the authors mention, saccades velocity can reflect reward value (see e.g. several cool papers by Reza Shadmehr's group on this topic, e.g. <https://doi.org/10.1523/JNEUROSCI.2621-15.2015>) and the other hand saccade response times can reflect also “certainty”, both as modulated by prior probability (e.g. seminal work by Carpenter & Williams, Nature 1995, <https://doi.org/10.1038/377059a0>) and the visibility of the target (for example manipulated by its luminance contrast – see for example Ludwig, Gilchrist & McSorley, 2004; <https://doi.org/10.1016/j.visres.2004.05.022>); so I think this statement should be changed

Response: Thank you for bringing this important point to our attention. We agree with you. We have changed the statement from “Given that reaction time and peak velocity reflect different features of perceptual decisions” to “it is unclear whether reaction time and peak velocity reflect different or similar features of perceptual decisions”. In the revised manuscript, we have also cited the new papers you introduced.

Main text, page 25, lines 521-522:

and on the other hand saccade reaction times can also reflect the degree of certainty with which a perceptual decision is made (i.e., confidence in a decision)^{40, 41}.

Main text, page 25, lines 525-527:

Although it is unclear whether reaction time and peak velocity reflect different or similar features of perceptual decisions,

Responses to evaluation of author's rebuttal to Reviewer 3 comments

1. **Reviewer's comment:** Approach & logic. As I mentioned above, I think the revised version of the paper has improved substantially in the clarity with which the research question is introduced and motivated, thanks to some rewording and additions. While I think the paper could still improve in terms of relating the present findings to the bigger picture of sensorimotor decision-making, I think overall the authors have addressed this point adequately.

Response: Thank you.

2. **Reviewer's comment:** Interpretation of results. The reviewer here raised a very important point, which I had missed in my own evaluation - that some participant's performance was at or near chance in the perceptual task. The authors replied providing additional analyses that shows how that effect they report are seen only (or nearly only) in participants who achieved 75% or more correct responses in the perceptual task. I think this additional analysis clarify the issues with interpretation of the results, although it remains unclear why a relatively large number of participants was at/near chance. Is it possible that these participants were prioritizing the movement task over the perceptual decision task, perhaps because they found it more challenging or engaging?"

Response: Yes, we think that it's possible. As you pointed out, those participants may have found it difficult to move their eye and/or hand quickly in the movement

task. For that reason, they may have prioritized the movement task over the perceptual decision task. We have added this description to the revised manuscript.

Main text, page 23, lines 470-472:

These participants may have prioritized the movement task over the perceptual decision task, perhaps because they found it difficult to move their eye and/or hand quickly in the movement task.

3. **Reviewer's comment:** In my opinion the authors have addressed adequately points 3 to 9.

Response: Thank you.

REVIEWERS' COMMENTS:

Reviewer #2 (Remarks to the Author):

The authors have done a thorough job in addressing all the remaining comments from the last round of revision. I have no further comments.